# Optimal and Scalable MAPF via
# Multi-Marginal Optimal Transport and Schrödinger Bridges

Usman A. Khan [1 2 3]   Joseph W. Durham [1]

## Abstract

We consider anonymous multi-agent path finding (MAPF) where a set of robots is tasked to travel to a set of targets on a finite, connected graph. We show that MAPF can be cast as a special class of multi-marginal optimal transport (MMOT) problems with an underlying Markovian structure, under which the exponentially large MMOT collapses to a linear program (LP) polynomial in size. Focusing on the anonymous setting, we establish conditions under which the corresponding LP is feasible, totally unimodular, and consequently, yields min-cost, integral ($\{0, 1\}$) transports that do not overlap in *both* space and time. To adapt the approach to large-scale problems, we cast the MAPF-MMOT in a probabilistic framework via Schrödinger bridges. Under standard assumptions, we show that the Schrödinger bridge formulation reduces to an entropic regularization of the corresponding MMOT that admits an iterative Sinkhorn-type solution. The Schrödinger bridge, being a probabilistic framework, provides a shadow (fractional) transport that we use as a template to solve a reduced LP and demonstrate that it results in near-optimal, integral transports at a significant reduction in complexity. Extensive experiments highlight the optimality and scalability of the proposed approaches.

## 1. Introduction

Coordinating large teams of robots to reach target locations while avoiding collisions in space and time is a fundamental problem in robotics and automation. In multi-agent path finding (MAPF), robots are assigned to targets on a shared graph and must compute collision-free trajectories that are jointly optimal in both space and time. This coupling of assignment, path planning, and scheduling renders MAPF combinatorial in nature. In this paper, we show that MAPF can be cast as multi-marginal optimal transport (MMOT) over path spaces with an underlying Markovian structure. Focusing on the anonymous setting, where any robot may reach any target, we show that the resulting Markovian MMOT admits a polynomial-size linear program (LP) with strong optimality and integrality guarantees (extensions to non-anonymous settings are possible via more general MMOT formulations). We then use ideas from the Schrödinger bridge framework and develop an entropic regularization of the corresponding MMOT to build scalable, probabilistic relaxations of the MAPF problem. The main contributions of this paper are as follows.

- We show that MAPF is a special class of Markovian MMOT, which admits a polynomial-size LP with a totally unimodular constraint matrix in the anonymous setting; subsequently, all extreme points of the feasible polyhedron are integral ($\{0, 1\}$). We further derive the conditions under which the proposed min-cost LP yields min-move and min-makespan transports.

- We connect the MAPF-MMOT to Schrödinger bridges, which enable transports with desirable structural properties through appropriate reference distributions. By choosing the reference as a Gibbs kernel, the Schrödinger bridge reduces to an entropic regularization of the MAPF-MMOT enabling Sinkhorn-type fast iterations. The resulting Sinkhorn-MAPF provides a *shadow transport*, a set of likely paths from robots to targets. As the Gibbs parameter $\varepsilon \to 0$, the Schrödinger bridge associated with a highly volatile reference Markov chain concentrates onto tight, min-cost geodesic corridors of the underlying graph.

- We use the shadow transport to guide a principled pruning of the underlying graph, resulting in an LP defined on fewer edges while preserving total unimodularity of the feasible polyhedron. Because the Schrödinger bridge leads to a shadow cast on highly likely paths,

[1]Amazon Robotics, Boston, MA, USA. [2]Boston College, Chestnut Hill, MA, USA. [3]UAK holds concurrent appointments as an Amazon Scholar with Amazon Robotics and as a Professor of Computer Science at Boston College. Correspondence to: Usman A. Khan <uakhan@amazon.com, usman.khan@bc.edu>.

*Proceedings of the 43rd International Conference on Machine Learning*, Seoul, South Korea. PMLR 306, 2026. Copyright 2026 by the author(s).

the pruned formulation retains near-optimal solutions while remaining scalable and integral. Experiments show that, as problem size grows, the resulting LP eliminates approximately 60–80% of the edges while incurring less than 10% cost degradation.

**Related Work:** Relevant MAPF overviews can be found in (Stern, 2019; Li et al., 2020); see (Agaskar et al., 2026) for a recent foundation model and also (Ma & Koenig, 2016; Peng et al., 2023; Ali & Yakovlev, 2023; Fine et al., 2023) for anonymous MAPF. Related algorithms include Conflict-Based Search, see e.g., (Sharon et al., 2015; Felner et al., 2017), and SAT-based solvers (Surynek, 2015). Prior work via flow networks and integer programs can be found in (Yu & LaValle, 2013; Ma, 2020); however, this line of work does not explicitly characterize the integrality of MAPF solutions as a polyhedral property of the associated LP. In contrast, our MAPF–MMOT formulation establishes integrality of the resulting LP from first principles via total unimodularity (Schrijver, 1986; Ahuja et al., 1993). The proposed MMOT viewpoint is novel and further enables extensions to probabilistic formulations. In particular, we generalize MAPF-MMOT to the Schrödinger bridge problem where the goal is to find a robot distribution $\mathbf{P}$ over path spaces that is close to a given reference $\mathbf{G}$. Key references in this direction include: (Léonard, 2014), which surveys the Schrödinger problem and its connections to OT; (Pavon & Ticozzi, 2010) on discrete-time Markovian bridges; and (Haasler et al., 2021) on tree-structured costs. MMOT and its computational aspects are discussed in (Pass, 2015; Lin et al., 2022; Haasler et al., 2023). Related work on multi-marginal Sinkhorn methods can be found in (Benamou et al., 2015; Di Marino & Gerolin, 2020; Carlier, 2022). Of relevance is also the body of work in (Chen et al., 2016; 2017; 2021) that casts Schrödinger bridges within a stochastic control framework.

## 2. Background and Problem Formulation

In this section, we provide some preliminaries and formally write the multi-agent path finding (MAPF) problem.

**Multi-Marginal OT:** Multi-marginal optimal transport (MMOT) extends the classical two-marginal OT problem to multiple marginals. The transport here is not between two measures but rather a coupling across more than two marginals. Formally, consider a joint probability tensor $\mathbf{P} \in \mathbb{R}^{K \times \cdots \times K}$, with $T + 1$ modes, each with $K$ supports, and a transport cost tensor $\mathbf{C}$ of the same dimensions. Given $T + 1$ marginals $\mathbf{q}_0, \mathbf{q}_1, \ldots, \mathbf{q}_T$, the MMOT problem is to find a min-cost transport plan (coupling), i.e.,

$$\min_{\mathbf{P} \geq 0} \langle \mathbf{P}, \mathbf{C} \rangle \quad \text{subject to} \quad \phi_t(\mathbf{P}) = \mathbf{q}_t, \forall t \in 0, \ldots, T,$$

such that all elements of $\mathbf{P}$ sum to 1, where $\langle \cdot, \cdot \rangle$ denotes

the inner product and $\phi_t$ is the (linear) projection of the joint distribution $\mathbf{P}$ on its $t$-th marginal, given to be $\mathbf{q}_t$. A generalization to compactly supported Borel measures on smooth manifolds can be found in (Pass, 2015). Clearly, the discrete MMOT problem is a linear program with $K^{T+1}$ variables in the transport tensor $\mathbf{P}$. It has been shown that for $T \geq 2$, combinatorial algorithms like the simplex methods no longer remain suitable (Lin et al., 2022). Because of the exponential state-space, recent work has studied convex relaxations, see e.g., (Lin et al., 2022; Haasler et al., 2023).

**From MMOT to MAPF:** We now establish a connection between multi-agent path finding (MAPF) and MMOT problems. Consider a finite, bounded region of interest $\Omega \subset \mathbb{R}^2$ in which $N$ robots operate. We discretize $\Omega$ into cells, each no smaller than a robot, and define a graph $\mathcal{G} = (\mathcal{V}, \mathcal{E})$ whose vertices $\mathcal{V}$ correspond to cells and whose edges $\mathcal{E}$ connect adjacent cells between which a robot can transition in one time step. The resulting graph is finite, and the cell size ensures that each vertex is occupied by at most one robot at any time. Let $|\mathcal{V}| = K$ and let $N$ robots in the set $\mathcal{N}$ occupy some vertices in $\mathcal{V}$ and travel on the edges $\mathcal{E}$, over a time horizon $T$. The goal for the robots is to reach a set $\mathcal{M}$ of $M$ distinct targets, also in $\mathcal{V}$, such that $M = N$, while minimizing the travel cost and/or the travel time to reach those targets[1]. This setup is standard in the discrete MAPF literature; see, e.g., (Standley, 2010; Yu & LaValle, 2013; Stern, 2019). We further let $\boldsymbol{\mu} = \{\mu_i\}$ and $\boldsymbol{\nu} = \{\nu_j\}$, both in $\mathbb{R}^K$, denote the initial distributions of $N$ robots and targets, respectively, on $K$ vertices.

Let $(X_t)_{t=0}^T$ be a discrete-time stochastic process with state-space $\{1, \ldots, K\}$, where each state corresponds to a vertex of the graph $\mathcal{G}$. In other words, $(X_t)_{t=0}^T$ is a possible trajectory taken by a robot over the time horizon $T$. Let $\mathbf{P} \in \mathbb{R}_{\geq 0}^{K \times \cdots \times K}$ be a $(T + 1)$-th order tensor representing a distribution on path space. In particular, each entry $\mathbf{P}_{i_0,\ldots,i_T}$ denotes the probability or the amount of mass (normalized to sum to one) assigned to a complete trajectory $X_0 = i_0, X_1 = i_1, \ldots, X_T = i_T$, that is, the path $i_0 \to i_1 \to \cdots \to i_T$ over the time horizon $t = 0, \ldots, T$, with $i_t \in \mathcal{V}$. Thus, $\mathbf{P}$ assigns mass directly to entire space-time trajectories rather than to individual vertices or edges. The collective motion of all $N$ robots, over the horizon $T$, is encoded in this joint path-space distribution and its one-time marginals $\mathbf{q}_t$ recover the robot locations at each time. We have $\sum_{i_0,\ldots,i_T} \mathbf{P}_{i_0,\ldots,i_T} = 1$ and

$$\sum_{i_0=1}^K \cdots \sum_{i_{t-1}=1}^K \sum_{i_{t+1}=1}^K \cdots \sum_{i_T=1}^K \mathbf{P}_{i_0,\ldots,i_T} = \frac{1}{N}[\mathbf{q}_t]_{i_t}, \quad (1)$$

with fixed endpoint distributions $\mathbf{q}_0 = \boldsymbol{\mu}$ and $\mathbf{q}_T = \boldsymbol{\nu}$.

---

[1]For simplicity, we assume that the number of robots and targets are equal. Extensions can be easily considered using unbalanced/partial OT but are beyond the scope of this exposition.

From the path-space tensor $\mathbf{P}$, the transport between consecutive time layers is obtained as a two-marginal. Specifically, the transport matrix $\Pi_t$ from time $t-1$ to $t$ is defined by

$$[\Pi_t]_{i_{t-1}, i_t} = N \sum_{i_0=1}^{K} \cdots \sum_{i_{t-2}=1}^{K} \sum_{i_{t+1}=1}^{K} \cdots \sum_{i_T=1}^{K} \mathbf{P}_{i_0, \ldots, i_T},$$

which represents the total mass (or joint probability) at vertex $i_{t-1}$ at time $t-1$ *and* at vertex $i_t$ at time $t$, aggregated over all past (before $t-1$) and future (after $t$) evolution. Clearly, since robot motion is causal in time, the process $(X_t)_{t=0}^{T}$ is Markovian, i.e., for all $t > 0$,

$$\mathbb{P}(X_t = u \mid X_{t-1} = v, \ldots, X_0 = w)$$
$$= \mathbb{P}(X_t = u \mid X_{t-1} = v),$$

for all $u, v, w \in \mathcal{V}$, and therefore the joint path-space tensor $\mathbf{P}$ admits the standard factorization:

$$\mathbf{P}_{i_0, \ldots, i_T} = \mathbb{P}(i_T \mid i_{T-1}) \cdot \ldots \cdot \mathbb{P}(i_1 \mid i_0) \cdot \mathbb{P}(i_0)$$
$$= \frac{1}{N} [\mathbf{q}_0]_{i_0} \prod_{t=1}^{T} \frac{\frac{1}{N} [\Pi_t]_{i_{t-1}, i_t}}{\frac{1}{N} [\mathbf{q}_{t-1}]_{i_{t-1}}}. \tag{2}$$

Given a cost tensor $\mathbf{C}$, with costs on each possible trajectory, the MAPF problem is to find $N$ robot trajectories to targets that minimize travel cost or travel time. In this paper, our interest is to exploit the aforementioned probabilistic interpretation of $\mathbf{P}$ and deploy MMOT and its efficient relaxations for MAPF. To our advantage, the Markovian factorization of the joint tensor reduces the number of free variables from $K^{T+1}$, exponential in $T$, to $K^2 T$, polynomial in $T$. Additionally, the general MMOT relaxes from fixing $T+1$ marginals to only two boundary marginals: the starting distribution $\mathbf{q}_0$, which is where the robots are located, and the ending distribution $\mathbf{q}_T$, where the targets are located. The intermediate marginals $\mathbf{q}_1, \ldots, \mathbf{q}_{T-1}$, consequently, define the positions the robots take when traveling from their starting locations to their destinations and are free under appropriate Markovianity/causality constraints as we will explicitly capture in the next section. This approach however comes at a price as MMOT, being a probabilistic object, does not necessarily result in integral $\frac{1}{N}\{0, 1\}$ transports. In other words, the transports may be fractional and the robots may split while traveling to the targets.

Building on this problem formulation, the remainder of the paper is organized as follows.

- Section 3 focuses on *integral* and *optimal* solutions of the proposed MAPF-MMOT. In particular, we show that the corresponding LP (**P1**) is totally unimodular under mild structural assumptions; consequently, all robots take non-conflicted, non-fractional paths to their targets. We further identify when the resulting min-cost paths also minimize moves and makespan.

- Section 4 builds towards scalability by casting MAPF as a probabilistic Schrödinger bridge and derives the corresponding entropic formulation and Sinkhorn-MAPF iterations (Appendix G). The main idea is to efficiently obtain a fractional shadow transport that concentrates mass on the highly likely paths.

- Section 5 then recovers integrality from the shadow transports by solving a variation of the base **P1** LP on a reduced graph obtained from the Schrödinger bridge, while Sections 6 and 7, respectively, provide complexity analysis and a comprehensive set of experiments.

## 3. Integral and Optimal Solutions for MAPF

Recall the problem formulation in Section 2 where $\boldsymbol{\mu}$ and $\boldsymbol{\nu}$ (robot and target locations) are such that

$$\mu_i = \begin{cases} 1, & i \in \mathcal{N}, \\ 0, & \text{otw.} \end{cases} \qquad \nu_j = \begin{cases} 1, & j \in \mathcal{M}, \\ 0, & \text{otw.} \end{cases}$$

with $\|\boldsymbol{\mu}\|_1 = \|\boldsymbol{\nu}\|_1 = N$. Note that the $(t-1) \to t$ transport plan $\Pi_t = \{\pi_{ij,t}\} \in \mathbb{R}^{K \times K}$ contains the robot distributions at time $t-1$ and $t$ as marginals, i.e.,

$$\mathbf{q}_{t-1} = \Pi_t \mathbf{1}, \qquad \mathbf{1}^\top \Pi_t = \mathbf{q}_t^\top, \qquad t = 1, \ldots, T,$$

where $\pi_{ij,t}$ is the amount of mass transported from vertex $i$, at time $t-1$, to vertex $j$, at time $t$. Our goal is to find the optimal transport plans $\{\Pi_t\}_{t=1}^{T}$ that move the robots (mass at source distribution $\boldsymbol{\mu}$) to the targets (mass at destination distribution $\boldsymbol{\nu}$) over the time horizon $T$. Recall that $\mathbf{C}$ is the cost tensor and let $C_t = \{c_{ij,t}\}$ denote the cost matrix at time $t$ such that the cost of traveling on an edge $i \to j$, starting at $t-1$ and arriving at $t$, is $c_{ij,t}$. We impose the following structural assumptions throughout this paper; see, e.g., (Standley, 2010; Yu & LaValle, 2013; Stern, 2019) for similar setups.

**Assumption 3.1.** The graph $\mathcal{G}$ and cost matrices $C_t$ satisfy the following:

(i) Self-loops $i \to i \in \mathcal{E}$, for all $i \in \mathcal{V}$, are always present. Consequently, waiting at any vertex is always feasible.

(ii) If two edges in $\mathcal{G}$ do not share a common endpoint, the corresponding physical motions can be executed simultaneously without conflict. In addition, an edge $i \to j$ is included in $\mathcal{G}$ only if its traversal is independent of the occupancy of all vertices other than $i$ and $j$.[2]

(iii) For each $i \to j \in \mathcal{E}$, the cost $c_{ij,t} < \infty, \forall t$, while $c_{ij,t} = +\infty, \forall (i, j) \notin \mathcal{E}$.

---

[2]This abstraction is a standard safeguard in cooperative path finding literature; see, e.g., (Standley, 2010). If needed, one may refine the graph so that collision-relevant interactions occur only at shared vertices; or on grid graphs, diagonal motion is disallowed.

(iv) We assume that $\mathbf{C}_{i_0,\ldots,i_T} = \sum_{t=1}^{T} c_{i_{t-1}i_t,t}$, i.e., the path cost is additive across time.

(v) The cost matrix satisfies the following, $\forall t = 1,\ldots,T$:

$$0 = c_{jj,t},\; j \in \mathcal{M} \;<\; c_{ii,t},\; i \notin \mathcal{M} \;<\; c_{ij,t},\; i \neq j.$$

The structure imposed in Assumptions 3.1(i)–(ii) ensures that the discretization faithfully captures the physical constraints of robot motion. Violating (i) or (ii) would mean that the graph does not correctly model the physical environment, and collisions invisible to the discretized formulation may arise. Assumption 3.1(iv) simply specializes the general path-space cost tensor to the standard stepwise cost used in MAPF, while 3.1(v) states that moving expends strictly more energy than waiting, and that only waiting at a target is free. We emphasize that Assumption 3.1 does not require grid graphs or point-mass robots. Any finite, connected $\mathcal{G}$ satisfying (i)-(ii) suffices, and the LP formulation and its guarantees developed in the sequel hold for any such $\mathcal{G}$.

With Assumption 3.1(iv), the tensor inner product $\langle \mathbf{P}, \mathbf{C} \rangle$ can be written as a sum over local transports and costs. Subsequently, because $\mathbf{P}$ is Markovian, the multi-marginal optimal transport formulation of anonymous MAPF can be equivalently written as follows:

$$\textbf{P1:} \qquad \{\Pi_t^*\}_{t=1}^{T} = \mathrm{argmin}_{\{\Pi_t\}_{t=1}^{T}} \sum_{t=1}^{T} \langle \Pi_t, C_t \rangle$$

$$\text{subject to} \quad \mathcal{F} := \begin{cases} \Pi_t \geq 0, \\ \Pi_t^\top \mathbf{1} = \Pi_{t+1}\mathbf{1}, \quad \forall t = 1,\ldots,T-1, \\ \Pi_1 \mathbf{1} = \boldsymbol{\mu},\; \Pi_T^\top \mathbf{1} = \boldsymbol{\nu}, \\ 0 \leq \Pi_t^\top \mathbf{1} \leq \mathbf{1}, \quad \forall t = 1,\ldots,T-1. \end{cases}$$

We explain this MMOT formulation next:

- **P1** is a linear program with linear constraints, in which the transport plans $\Pi_t$'s are real-valued, nonnegative decision variables (not necessarily integral).

- Gluing constraints: The second set of equality constraints impose a Markovian restriction to the global MMOT problem. They ensure that the mass transported from $t$ to $t+1$, must be the mass that arrives at $t$ from $t-1$, i.e., $\Pi_{t+1}\mathbf{1} = \mathbf{q}_t$ and also $\Pi_t^\top \mathbf{1} = \mathbf{q}_t$.

- Terminal constraints: The boundary marginals are fixed to enforce the robots' initial and terminal positions.

- Vertex-capacity constraints: The last inequality constraints ensure that no location (vertex in $\mathcal{G}$) receives more than one robot.

Clearly, if $\Pi_t$'s are integral, i.e., $\pi_{ij,t} \in \{0,1\}, \forall i,j,t$, then we get executable robot paths to the targets. Assumption 3.1

and the constraint polyhedron $\mathcal{F}$ further ensure that all targets are reached, while the robot trajectories are non-conflicting and min-cost. We characterize these results next.

### 3.1. Main Results

We now characterize some basic properties of **P1** in the following lemmas.

**Lemma 3.2.** *Let $\mathcal{G} = (\mathcal{V}, \mathcal{E})$ be a finite, connected graph with $|\mathcal{V}| = K$. Consider $N$ robots in $\mathcal{N}$ and $M$ targets in $\mathcal{M}$, on distinct vertices in $\mathcal{G}$, such that $N = M \leq K/2$. For each time $t \in \mathbb{N}$, let $C_t = \{c_{i,j,t}\}$ satisfy Assumption 3.1. Then, there exist a finite $\bar{T} \in \mathbb{N}$ and a transport $\{\Pi_t\}_{t=1}^{\bar{T}}$ feasible for P1, i.e., satisfying all constraints in $\mathcal{F}$, such that $\sum_{t=1}^{\bar{T}} \langle \Pi_t, C_t \rangle < \infty$.*

The proof is provided in Appendix A. Lemma 3.2 establishes the feasibility of anonymous MAPF and the constraint set $\mathcal{F}$ and does so purely at a structural level, without invoking any properties of the corresponding linear program (LP) beyond Assumption 3.1. The following lemma now concretely establishes the properties of the LP in **P1**.

**Lemma 3.3.** *Consider the settings of Lemma 3.2 and Assumption 3.1, and fix a horizon $\bar{T}$ such that $\mathcal{F}$ is nonempty with $\sum_{t=1}^{\bar{T}} \langle \Pi_t, C_t \rangle < \infty$, for some feasible $\{\Pi_t\}_t$. Then, the MMOT formulation P1 over the horizon $\bar{T}$ satisfies:*

- (i) *P1 admits an optimal solution $\{\Pi_t^*\}_{t=1}^{\bar{T}}$ that is integral, i.e., $\Pi_t^* \in \{0,1\}^{K \times K}$, for all $t = 1,\ldots,\bar{T}$.*

- (ii) *P1 results in a transport $\{\Pi_t^*\}_{t=1}^{\bar{T}}$ that attains a minimum cost over the horizon $\bar{T}$.*

- (iii) *The complexity of P1 is polynomial in $K$ and $\bar{T}$.*

The proof is provided in Appendix B, where we show that **P1** admits an optimal *integral* basic solution, as a consequence of total unimodularity of the constraint matrix (Schrijver, 1986). The result can also be viewed through the lens of classical integrality arguments for network flows (Ford & Fulkerson, 1962; Ahuja et al., 1993). Regarding (iii), see Section 6 for precise complexity arguments and Section 7 for related empirical results. The next theorem uses the results of the previous two lemmas and applies them to the anonymous MAPF problem.

**Theorem 3.4.** *Consider the settings of Lemmas 3.2, 3.3, and Assumption 3.1. The optimal transport plan $\{\Pi^*\}_{t=1}^{\bar{T}}$, returned by P1, satisfies:*

- (i) *No two robots collide at any time.*

- (ii) *The robot trajectories do not overlap in both space and time.*

- (iii) *All robots reach a distinct target.*

Theorem 3.4 is one of the central results of this paper; its proof and some important insights are provided in Appendix C. It establishes the conditions under which MMOT gives executable trajectories, i.e., $\{0,1\}$ transports, relying on the total unimodularity (TU) of **P1**. TU in general is delicate, and adding arbitrary constraints to enforce a desirable behavior typically breaks TU. It is therefore preferable to impose desirable traits in the trajectories through the cost matrices $\{C_t\}_t$. We next describe a few such properties, in addition to Assumption 3.1; it can be verified that they do not violate Lemma 3.3 and Theorem 3.4.

**No oscillations**: For all distinct vertices $i \neq j \in \mathcal{V}$, and all $t$, let $c_{ii,t} + c_{ii,t+1} < c_{ij,t} + c_{ji,t+1}$. In other words, leaving $i$ and returning to it, now or later, costs more than staying at $i$.
**Temporal urgency**: For all $i \to j \in \mathcal{E}$, all $t$, and all feasible moves, let $c_{ij,t} \leq c_{ij,t+1}$. Thus, executing a move is never cheaper later. Combined with Assumption 3.1(v), this implies that whenever a robot can reach a target earlier, there exists an optimal transport in which it does so and subsequently waits at the target.
**Temporal subadditivity**: For all $i, j, k \in \mathcal{V}$ and all $t$, let $c_{ij,t} \leq c_{ik,t} + c_{kj,t+1}$. Thus, whenever a direct move from $i$ to $j$ is available, routing through an intermediate vertex over consecutive time steps is never cheaper. This rules out avoidable detours: if a robot can move directly to a vertex and then wait, it is never optimal to reach the same vertex via an unnecessary intermediate location.
The two temporal conditions described above should not be used in scenarios, e.g., where a toll road becomes cheaper later (within the horizon $\bar{T}$) and the goal is to exploit that.
**Shortest-path costs**: Assume that $\mathcal{G}$ is endowed with an edge-length metric $d : \mathcal{E} \to \mathbb{R}_+$, such that for all $i \to j \in \mathcal{E}$ and all $t$, $c_{ij,t} = d(i,j)$. Under this cost structure, any minimum-cost transport minimizes the total traveled distance among all feasible transports. A canonical example is robots and targets embedded in $\mathbb{R}^2$, with $d(i,j)$ given by the Euclidean distance.

**Minimum Moves:** It can be shown that given a feasible horizon $\bar{T}$, under transition costs $c_{i \neq j,t} = 1$, for all $t$, and for sufficiently small waiting costs $c_{ii,t} > 0$, for $i \notin \mathcal{M}$, a min-cost, integral solution of **P1** is also a min-move solution.

**Minimum Makespan:** For any feasible transport $\{\Pi_t\}_{t=1}^{\bar{T}}$, its makespan is the largest time-index for which $\pi_{ij,t} > 0$, for some $i \neq j$. The minimum makespan is the smallest such value over all feasible transports. We next describe obtaining a minimum makespan transport from **P1** by tuning costs with the help of the following assumption and lemma.

**Assumption 3.5.** Let $\{\tilde{c}_{ij}\}_{i \to j \in \mathcal{E}}$ satisfy $\tilde{c}_{jj} = 0$, $j \in \mathcal{M}$, and $0 < \tilde{c}_{ii,i \notin \mathcal{M}} < \tilde{c}_{ij,i \neq j}$ and define $\tilde{c}_{\min}$ to be the minimum over all $\tilde{c}_{ij}$, with $i \neq j$. Choose $C_t = \{c_{ij,t}\}$ to be such that $c_{ij,t} := B^t \tilde{c}_{ij}, \forall i \to j \in \mathcal{E}, \forall t$, where $B$ is chosen to satisfy $(B-1)\tilde{c}_{\min} > \sum_{i \to j \in \mathcal{E}} \tilde{c}_{ij}$.

Note that Assumption 3.5 imposes a much stronger growth condition on the costs. It can be further verified that it satisfies Assumption 3.1 and also implies the aforementioned no-oscillations and temporal cost conditions.

**Lemma 3.6.** *Let $\mathcal{G}$ be finite and connected and consider $C_t$ such that it satisfies Assumption 3.5. Let $\bar{T}$ be a feasible horizon for **P1**. Then, for any optimal solution $\{\Pi_t^*\}_{t=1}^{\bar{T}}$ of **P1**, all robot motions terminate by $T^*$, where $T^*$ is the minimum makespan over all feasible transports.*

The proof is provided in Appendix D. Note that Lemma 3.6 provides a transport over the entire horizon $\bar{T}$ and achieving minimum makespan is implicit. In other words, the robot motion ceases at $T^*$ because of rapidly growing time-dependent costs. Such costs may cause numerical instability; to avoid that the following result explicitly searches for the minimum feasible horizon $T^*$.

**Lemma 3.7.** *For a feasible anonymous MAPF instance over a finite, connected graph $\mathcal{G}$ with $|\mathcal{V}| = K$, the minimum makespan satisfies $T^* \leq N + K - 1$. A minimum makespan transport can be found in $\mathcal{O}(\log K)$ calls to **P1**.*

The $T^*$ bound in this lemma can be found, e.g., in (Yu & LaValle, 2013; Ma, 2020). The rest of the lemma follows by performing a binary search over the horizon $T \in [0, N + K - 1]$ and checking for the earliest feasibility of **P1**. We thus obtain two complementary approaches for computing minimum makespan transports in Lemmas 3.6 and 3.7. The exponential cost construction in Assumption 3.5 and consequently Lemma 3.6 implicitly encode the makespan optimality into the objective, at the expense of aggressive cost scaling. Alternatively, minimum makespan can be found explicitly by searching over the horizon, requiring $\mathcal{O}(\log K)$ calls to **P1** with simpler cost.

An alternate min-makespan formulation can be achieved by minimizing $z$, such that $z \geq t \pi_{ij,t}, \forall i \notin \mathcal{M}, j \in \mathcal{V}, t$. However, the resulting LP is not totally unimodular in general. In other words, explicit makespan minimization requires integer programs, and therefore the implicit mechanism described here may be preferable.

The results in this section yield a polynomial-time LP for anonymous MAPF; see also Section 6. This complexity however may be impractical for very large-scale MAPF. In the subsequent sections, we develop scalable solutions for **P1** by casting MAPF as a discrete Schrödinger bridge, which provides a principled probabilistic framework for formulating and analyzing MAPF (Section 4). This formulation, under appropriate conditions, leads to a convex Problem **P2**, which we show is an entropic relaxation of **P1**. The resulting **P2** admits highly efficient Sinkhorn-type iterations (Appendix G) but yields fractional (shadow) transports. Integrality is then enforced with a subsequent projection step to recover executable robot motions (Section 5).

# 4. MAPF and the Schrödinger Bridge Problem

The classical Schrödinger bridge is described as follows. Given a reference diffusion process $G$ on a topological state space $\mathcal{X}$, with arbitrary marginals, find a probability measure $P^*$ on the space of continuous trajectories that minimizes the relative entropy with respect to $G$:

$$P^* = \operatorname{argmin}_P \operatorname{KL}(P \,\|\, G),$$

such that the measure $P^*$ has fixed initial and terminal marginals (Schrödinger, 1931; Léonard, 2014). Here, $G$ is typically chosen as Brownian motion and $P^*$ is the most plausible stochastic evolution whose continuous-time trajectories interpolate between the given boundary marginals. Schrödinger bridge formulations are widely used in applied physics and are typically studied in continuous settings to model the evolution of particle systems, see e.g., the hot and lazy gas experiments in (Villani, 2009; Léonard, 2017). Originally introduced by Erwin Schrödinger in the 1930s, the Schrödinger bridge problem characterizes the path-space trajectories of gas particles from empirical observations of their distributions at two time instants, and is closely connected to large deviation theory, where Schrödinger bridges arise as minimizers of associated rate functionals (Föllmer, 1988). We now cast MAPF as a Schrödinger bridge and characterize the conditions under which this formulation reduces to the MAPF–MMOT problem **P1**.

For the remainder of the paper, we assume that $T < \infty$ is a feasible horizon for **P1**. Recall the factorization of $\mathbf{P}$ in (2) and let $\mathbf{G}$ denote a reference Markovian tensor on $\mathcal{V}$ with a similar factorization, where $\mathbf{G}_t$ are the transports of the reference distribution $\mathbf{G}$ with marginals $\mathbf{G}_t \mathbf{1} = \mathbf{g}_{t-1}$ and $\mathbf{G}_t^\top \mathbf{1} = \mathbf{g}_t$. The Schrödinger bridge problem corresponding to **P1** seeks a joint distribution $\mathbf{P}$, and the corresponding sequence of transports $\{\Pi_t\}_{t=1}^T$, that is closest to $\mathbf{G}$ in the relative entropy sense, i.e.,

$$\min_{\mathbf{P} \in \mathcal{C}} \operatorname{KL}(\mathbf{P} \,\|\, \mathbf{G}), \qquad (3)$$

where $\mathcal{C}$ is the set of tensors satisfying the constraints of **P1** and each element of $\mathbf{G}$ is such that the KL divergence is well-defined.

**Lemma 4.1.** *The Schrödinger bridge in* (3) *reduces to*

$$\operatorname{KL}(\mathbf{P} \,\|\, \mathbf{G}) = \sum_{t=1}^T \operatorname{KL}\left(\tfrac{1}{N}\Pi_t \,\|\, \mathbf{G}_t\right) + \operatorname{KL}\left(\tfrac{1}{N}\mathbf{q}_0 \,\|\, \mathbf{g}_0\right)$$
$$- \frac{1}{N}\sum_{t=1}^T \operatorname{KL}\left(\tfrac{1}{N}\mathbf{q}_{t-1} \,\|\, \mathbf{g}_{t-1}\right). \qquad (4)$$

The proof is provided in Appendix E. Note that the full Schrödinger bridge formulation decomposes into transport and marginal KL terms (4); a related decomposition is derived in (Pavon & Ticozzi, 2010) for the initial-final marginal problem using conditional transition probabilities. Eq. (4) represents the general form of the Schrödinger bridge, through which one may impose a desirable structure on the robot trajectories by appropriately choosing the reference transports $\mathbf{G}_t$ and marginals $\mathbf{g}_t$. Consequently, (4) returns a MAPF transport $\mathbf{P}$ that is consistent with the initial and final robot locations while remaining close to $\mathbf{G}$. Existence and uniqueness of Schrödinger bridges are studied in (Pavon & Ticozzi, 2010), where a solution is derived using space-time harmonic functions under suitable assumptions on the reference processes. In the following, we restrict the reference distributions to the Gibbs form, which leads to Sinkhorn-type iterations.

**Lemma 4.2.** *Let $C_t = \{c_{ij,t}\}$ be the cost matrix and let $\mathbf{G}_t = \{\bar{g}_{ij,t}\}$ be the normalized Gibbs kernel, i.e., $\bar{g}_{ij,t} := \frac{g_{ij,t}}{z_t}, g_{ij,t} := \exp\left(-\frac{c_{ij,t}}{\varepsilon}\right),$ and $z_t := \sum_{k,\ell} g_{k\ell,t}.$ Then, for each $t$,*

$$\operatorname{KL}\left(\tfrac{1}{N}\Pi_t \,\|\, \mathbf{G}_t\right) = \frac{1}{N}\sum_{i,j} \pi_{ij,t} \log \pi_{ij,t}$$
$$+ \frac{1}{N}\sum_{i,j} \pi_{ij,t} \frac{c_{ij,t}}{\varepsilon} + \log \tfrac{1}{N} + \log z_t.$$

The proof is provided in Appendix F. The above lemma leads to the following Schrödinger bridge formulation of MAPF, when the reference distribution $\mathbf{G}$ is the Gibbs kernel (after removing the constants that do not depend on the transport variables) and minimizing $\varepsilon N \operatorname{KL}(\tfrac{1}{N}\Pi_t \| \mathbf{G}_t)$:

$$\textbf{P2:} \quad \min_{\{\Pi_t \in \mathcal{F}\}_{t=1}^T} \sum_{t=1}^T \left( \langle \Pi_t, C_t \rangle + \varepsilon \sum_{i,j} \pi_{ij,t}(\log \pi_{ij,t} - 1) \right)$$

under the constraints of **P1**. We let $\{\widetilde{\Pi}_t\}_{t=1}^T$ denote the transport returned by **P2** and note that **P2** is precisely the entropic regularization of **P1**.

To obtain efficient and scalable solutions, **P2** imposes the non-negativity constraints on the transport variables and allows $\Pi_t \geq 0$. This relaxation yields a convex problem that can be solved efficiently using Sinkhorn-type iterations. Under this relaxation, the marginal distributions induced by $\Pi_t$ may become fractional, and the marginal KL terms in the full Schrödinger bridge objective (Eq. (4)) may not remain constant. Consequently, **P2** may no longer remain exactly equivalent to the Schrödinger bridge but can be interpreted as a tractable relaxation thereof. Solving the convex relaxation **P2** yields fractional (shadow) transports, over which integrality can be imposed, as we will develop in Section 5. We provide the corresponding multi-marginal Sinkhorn iterations to build the fractional shadow transports $\widetilde{\Pi}$ in Appendix G. A formal discussion and analysis of Sinkhorn-MAPF is beyond the scope of this paper.

## 5. Integral Projection of Sinkhorn-MAPF

To obtain an integral solution from the entropy-regularized transports $\{\widetilde{\Pi}_t\}_{t=1}^T$, obtained from **P2**, we project $\widetilde{\Pi}_t$'s back on the totally unimodular polyhedron $\mathcal{F}$. To this end, we minimize a modified objective $\langle \Pi_t, C_t\rangle + \lambda \operatorname{KL}(\Pi_t\|\widetilde{\Pi}_t)$ that penalizes the transport $\Pi_t$ when it's far from $\widetilde{\Pi}_t$. Adding the KL penalty however makes the objective non-linear, which we address by linearizing around an operating point $\pi_{ij,t}^0$, yielding

$$\operatorname{KL}(\Pi_t\|\widetilde{\Pi}_t) \approx \sum_{i,j} \pi_{ij,t}\left(\log \pi_{ij,t}^0 - \log \widetilde{\pi}_{ij,t}\right)$$
$$+ \sum_{i,j} \pi_{ij,t} - \sum_{i,j} \pi_{ij,t}^0.$$

Dropping constants and choosing $\pi_{ij,t}^0$ to be a constant over all $i,j$, we obtain

**P3:** $\quad \min_{\{\Pi_t\}_{t=1}^T} \sum_{t=1}^T \left(\sum_{i,j} \pi_{ij,t}\left(c_{ij,t} - \lambda \log(\widetilde{\pi}_{ij,t} + \delta)\right)\right)$

$\quad$ subject to $\quad \Pi_t \in \mathcal{F},\ \Pi_t \subseteq [\widetilde{\Pi}_t]_\eta, \forall t,$

where $\delta \geq 0$ ensures the logarithm is well-defined, and $[\widetilde{\Pi}_t]_\eta$ is the regularized transport $\widetilde{\Pi}_t$ with zeros for all elements that are at most $\eta$. We let $\{\widehat{\Pi}_t\}_{t=1}^T$ denote the transport obtained from **P3**, which is effectively built from the *shadow* transport **P2**, i.e., the entropic regularization of **P1**. We provide a few important remarks regarding **P3** and its parameters next:

- The constants $\lambda, \delta$ must be chosen carefully to ensure that the modified cost $c_{ij,t} - \lambda \log(\widetilde{\pi}_{ij,t} + \delta)$ lies in the purview of Assumption 3.1. Clearly, $\{\widehat{\Pi}_t\}_t$ is integral $\{0, 1\}$, because **P3** is an LP under the same feasibility polyhedron $\mathcal{F}$ of **P1**. However, the pruned graph or $T$ may no longer remain feasible and $\eta$ may need to be adjusted accordingly.

- The interplay between the three constants $\varepsilon, \lambda, \eta$ straddles the spectrum of transports from optimal to highly scalable. Note that $\lambda = \eta = 0$, disconnects **P2** (regardless of $\varepsilon$) and recovers **P1**.

- Choosing $\varepsilon, \lambda > 0$, independent of $\eta$, biases **P3** to place mass on edges with large $\widetilde{\pi}_{i,j,t}$, since the modified objective in **P3** arises from a linearized KL divergence between $\widehat{\pi}_{i,j,t}$ and $\widetilde{\pi}_{i,j,t}$. Consequently, $\widehat{\pi}_{i,j,t}$ may inherit the smoothing effect induced by **P2** as $\varepsilon \uparrow$.

- A scalable recipe is apparent: solve convex **P2** fast, build a shadow, and subsequently solve the **P3** LP over a pruned graph (with appropriate choices of $\varepsilon, \eta, \lambda, \delta$). A detailed breakdown of these parameters, based on 260 experiments on a 1.5M-variable problem, is provided in Appendix H.3.

## 6. Computational Complexity

The LP in Problem **P1** returns $T$ transport matrices $\{\Pi_t\}_{t=1}^T$, each supported on $|\mathcal{E}|$ edges, resulting in $n := |\mathcal{E}|T$ decision variables and $\mathcal{O}(n)$ constraints. On physical graphs with bounded-degree connectivity (e.g., nearest-neighbor motion), $|\mathcal{E}| = \mathcal{O}(K)$ and thus $n = \mathcal{O}(KT)$. Using classical interior-point methods for linear programming, Problem **P1** admits polynomial worst-case bit-complexity, for example $\mathcal{O}(n^3 L)$, where $L$ is the encoding length of the input data (Karmarkar, 1984; Goldfarb & Todd, 1989). The corresponding methods may not return a basic optimal solution (a vertex of the constraint polyhedron); in such cases, standard crossover techniques can be used to recover an extreme-point solution (Wright, 1997; Potra & Wright, 2000). While these methods provide polynomial-time guarantees, practical implementations often rely on simplex-based methods, which perform significantly faster on large-scale instances despite lacking polynomial worst-case guarantees.

To improve scalability, **P2** solves an entropic relaxation via Sinkhorn iterations (Appendix G) that are specialized to the MAPF problem and its constraints. The convergence and complexity of multi-marginal Sinkhorn are studied, e.g., in (Di Marino & Gerolin, 2020; Carlier, 2022), where linear convergence is established when the iterations are viewed as block coordinate descent on a convex dual objective. In practice, only a small number of Sinkhorn iterations suffices to construct the pruned graph used later in **P3**, as demonstrated in Section 7 and in the detailed experiments (Appendix H). Finally, Problem **P3** solves the original linear program over the pruned graph and therefore has the same worst-case complexity bound as the aforementioned interior-point methods. However, due to the effective graph pruning based on the shadow transport of **P2**, the number of variables is significantly reduced, with $n \approx \zeta |\mathcal{E}|T$, where in practice $\zeta$ typically lies in the range $[0.2, 0.4]$ for large-scale instances, as demonstrated in the experiments.

The above complexity bounds are worst-case limits. In practice, modern LP solvers such as Gurobi are highly optimized and we observe empirical solve time scaling, with fixed $T = 30$, as $O(K^{1.68})$ for **P1** optimal and as $O(K^{1.15})$ for the scalable **P2+P3** pipeline; see Fig. 4 and Appendix H.2 for a detailed scaling study.

## 7. Experiments

We consider a grid of size $K = W \times H$ with potential obstacles; while the framework applies to arbitrary graphs, grids are chosen for simplicity. Motion is allowed in the cardinal directions, while diagonal motion is prohibited. The waiting cost is $0$ at targets and $0.5$ at non-targets; move cost is $c_{i \neq j} = 1$, for all $i \to j \in \mathcal{E}$. The experiments in Figs. 1–3 are conducted using the HiGHS solver from the SciPy lin-

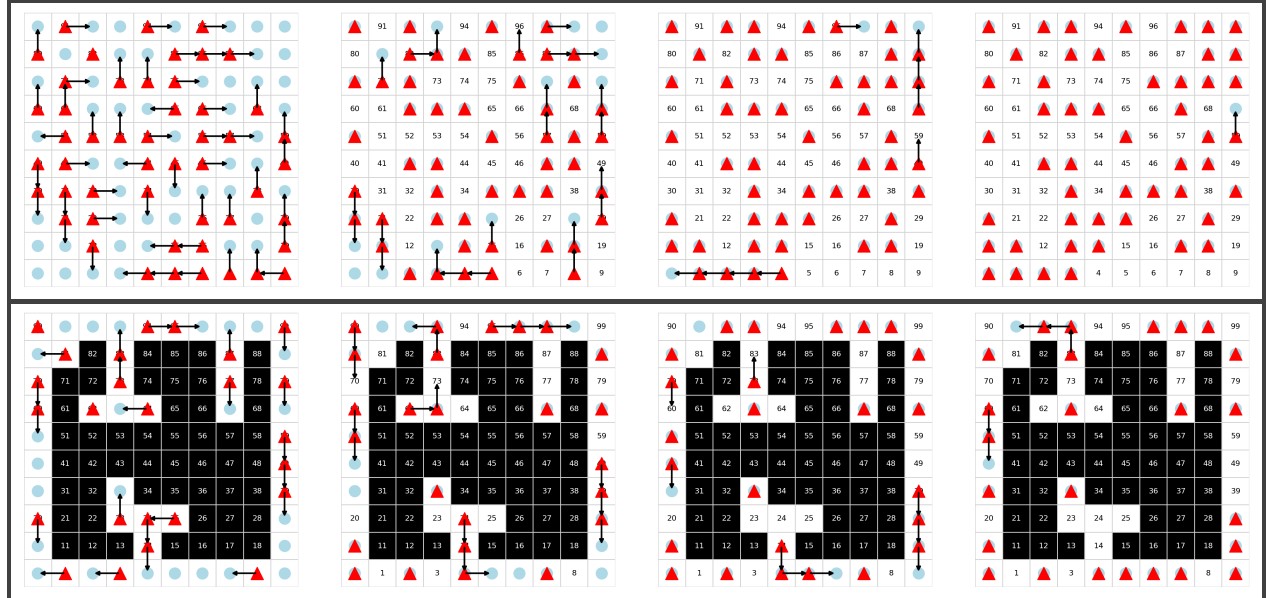

*Figure 1.* A $10 \times 10$ grid with $\{50, 25\}$ robots ▲ and targets ●, and $\{0, 50\}$ obstacles. From left to right: $t = 0, 1, 2, 3$.

prog library. Fig. 1 shows the min-cost optimal **P1** transport for two arbitrary configurations over $T = 3$; we note that arbitrarily placed obstacles alter the connectivity and diameter of the underlying graph, and it no longer remains a regular grid. We next demonstrate the Schrödinger shadow transport (**P2**) in Fig. 2 on a $40 \times 40$ grid with $\{20, 80\}$ robots and targets; all trajectories are superimposed. The left figures show the optimal integral transports **P1**; the middle figures show the (fractional) Sinkhorn-MAPF with $\varepsilon = 50$; while the right figures show integral **P3** transports obtained on pruned graphs with nominal cost degradation. Finally, Fig. 3 demonstrates the cost degradation versus edges kept from the shadow transport on a $K = W^2$ grid with $2W$ robots. We observe that shadow-based pruning is much more effective and feasibility is obtained with a smaller number of edges as $K \uparrow$ with a nominal cost degradation.

To evaluate the runtime scaling, we switch to the Gurobi LP solver and conduct 162 independent runs on square grids ranging from $K = 2,500$ to $K = 22,500$ vertices at $5\%$ robot density ($N = 0.05K$) and $T = 30$. Fig. 4 shows that the optimal **P1** solve time grows as $O(K^{1.68})$, while the **P2+P3** pipeline scales nearly linearly as $O(K^{1.15})$, yielding speedups from $3.6\times$ to $7.1\times$ with cost gap consistently below $10\%$. Every solution across all runs is verified integral. See Appendix H.2 for the full scaling study.

Table 1 reports the average cost gap (%) of the **P2+P3** pipeline over 260 runs at $K = 10,000$ for varying $(\varepsilon, \lambda)$. We observe that the regularization parameter $\varepsilon$ is the dominant factor: small $\varepsilon$ produces a concentrated shadow close to the **P1** optimum, while $\lambda$ has a milder effect. A robust default is $\varepsilon = 0.2, \lambda = 0$: a $4.3\%$ gap at $5\times$ speedup. See

Appendix H.3 for the full sensitivity analysis.

*Table 1.* Average cost gap (%) relative to **P1** optimal for each $(\varepsilon, \lambda)$ pair, over 13 instances at $K = 10,000$, $T = 30$.

| $\varepsilon \setminus \lambda$ | 0 | 0.5 | 1.0 | 5.0 |
|---|---|---|---|---|
| 0.1 | 2.3 | 2.5 | 2.7 | 3.0 |
| 0.2 | 4.3 | 5.0 | 5.8 | 7.3 |
| 0.5 | 11.1 | 12.7 | 14.0 | 16.5 |
| 1.0 | 17.3 | 18.9 | 20.1 | 23.1 |
| 5.0 | 17.1 | 18.0 | 18.7 | 19.9 |

Appendix H.4 validates the proposed approaches under non-uniform costs, and Appendix H.5 compares against the CBM baseline of (Ma & Koenig, 2016).

## 8. Conclusions

In this paper, we develop a principled framework for multi-agent path finding (MAPF) that bridges multi-marginal optimal transport, entropy-regularized relaxations, and linear programming. By showing total unimodularity of the feasibility polyhedron in the anonymous case, we obtain integral transports in polynomial time without explicitly enforcing integrality. We extend the methodology to Schrödinger bridges and entropic formulations that provide a novel probabilistic viewpoint of MAPF, yielding scalable approximations and structural guidance for reducing problem size. Building on this structure, the proposed projection and pruning strategies enable efficient recovery of executable transports, offering a flexible trade-off between tractability, smoothness, and optimality.

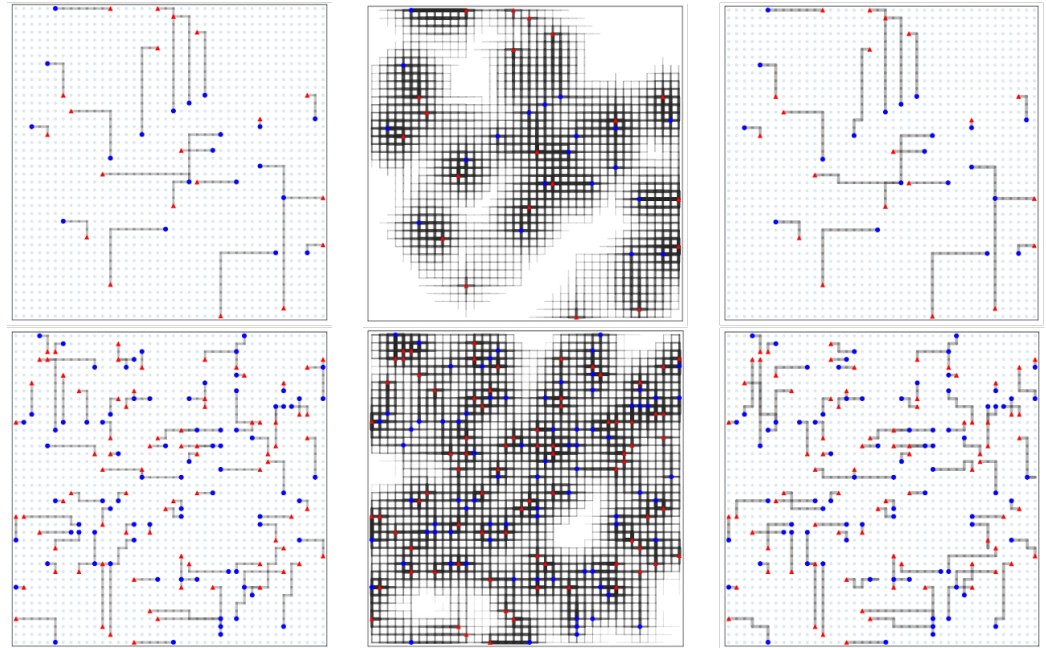

*Figure 2.* (Left) Optimal **P1**; (Middle) Schrödinger shadow transport **P2**; (Right) Integral projection **P3**. Top ($N = 20, T = 15$): **P1** cost 181; **P2** cost 1053; **P3** with 23% edges retained at cost 181, i.e., 0% degradation. Bottom ($N = 80, T = 10$): **P1** cost 402; **P2** cost 3160; **P3** with 22% edges retained at cost 436, i.e., 8.5% degradation.

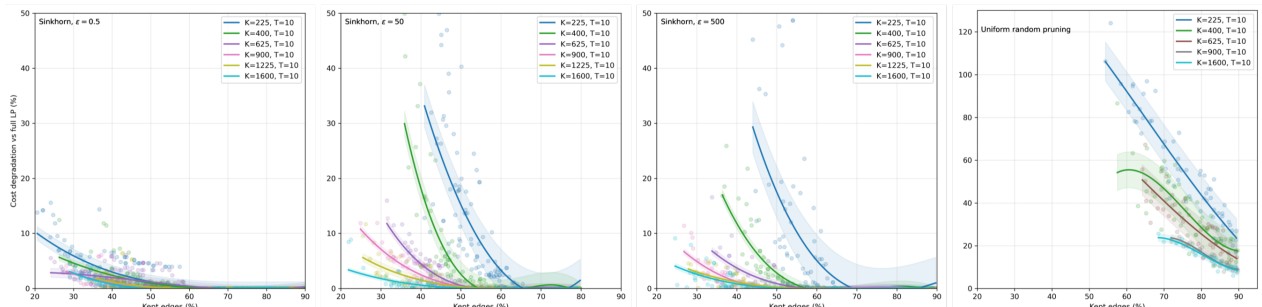

*Figure 3.* Scalable MAPF: The vertical axis plots the cost degradation $x$, i.e., the cost of transport obtained from **P3** is $(1 + x)c_{opt}$, where $c_{opt}$ is the optimal min-cost from **P1**; the horizontal axis plots the % of edges retained from the full **P1** transport.

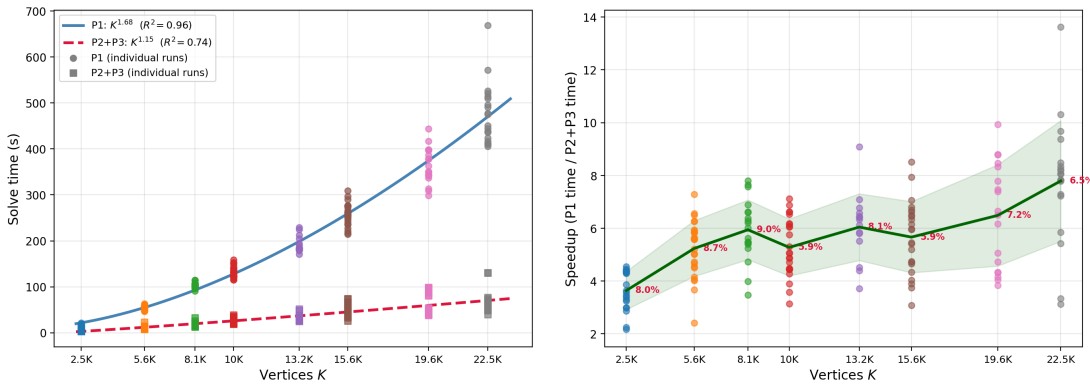

*Figure 4.* Runtime scaling across 162 runs at 5% robot density, $T = 30$. (Left) Solve time for **P1** optimal (circles) and **P2+P3** (squares) versus $K$; curves show power-law fits $aK^p + b$. (Right) Speedup versus $K$; the green line connects averages, the shaded band shows $\pm 1$ std. dev., and red annotations indicate the average cost gap.

## Impact Statement

This paper presents work whose goal is to advance the field of Machine Learning and Robotics. There are many potential societal consequences of our work, none of which we feel must be specifically highlighted here.

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

## Appendix Overview

The appendix is organized as follows.

## A. Proof of Lemma 3.2

*Proof.* Consider a Markov chain $\mathrm{M}$ whose state-space consists of all $K$-dimensional $\{0,1\}$ vectors with *exactly* $N$ ones, representing the configuration space of the robots on the graph $\mathcal{G}$. A transition from state $\mathbf{q} \in \mathrm{M}$ to $\mathbf{q}' \in \mathrm{M}$ is allowed (and given a non-zero probability), whenever $\mathbf{q}'$ is obtained from $\mathbf{q}$ by moving a single robot along a traversable edge $(i,j) \in \mathcal{E}$ to a neighboring location $j$ that is unoccupied by a robot, or by keeping all robots at their current vertices. In other words, the state $\mathbf{q}$ has outgoing (positive probability) edges to every state that can be achieved with exactly one robot's valid move and also to itself (no robot moves). Because $\mathcal{G}$ is connected, the robots are indistinguishable, a standard result from the pebble motion literature states that the configuration space of $\mathrm{M}$ is connected (Kornhauser et al., 1984). Additionally, since $\mathcal{G}$ is finite, under the imposed transition probabilities, $\mathrm{M}$ is irreducible and every state in $\mathrm{M}$ is recurrent, i.e., from any state $\mathbf{q}_1 \in \mathrm{M}$, one can reach any other state $\mathbf{q}_2 \in \mathrm{M}$, by a finite sequence of single-robot moves along the edges of $\mathcal{G}$, with probability 1. Therefore, the terminal configuration $\boldsymbol{\nu}$ is reachable from $\boldsymbol{\mu}$ in a finite number of steps $\bar{T}$. Clearly, a transport sequence $\{\Pi_t\}_{t=1}^{\bar{T}}$ that encodes these Markovian transitions satisfies all feasibility constraints in $\mathcal{F}$, and the lemma follows, since $C_t$ is finite on $\mathcal{E}$. $\qquad\square$

## B. Proof of Lemma 3.3

*Proof.* We first show (i): In order to establish the integral solution guarantee by the LP in **P1**, we show that all constraints in $\mathcal{F}$ can be written in the form of a node-arc incidence matrix of an augmented directed graph $\bar{\mathcal{G}}$ that we construct as follows. Consider the time-expanded graph $\mathcal{G}_0^{\bar{T}}$ over the horizon $\bar{T}$, i.e., with vertices $(i,t)$, for each $i \in \mathcal{V}$ and time $t = 0, \ldots, \bar{T}$, and arcs $(i, t-1) \to (j, t)$, for $t = 1, \ldots, \bar{T}$, whenever the move $i \to j$ is allowed by $\mathcal{G}$. We further apply the classical node-splitting construction of Ford and Fulkerson (Ford & Fulkerson, 1962; Ahuja et al., 1993) to augment $\mathcal{G}_0^{\bar{T}}$ as follows. For every intermediate vertex $(i,t), t = 1, \ldots, \bar{T} - 1$, split it into two vertices

$$(i,t)'' \quad \text{(in-node)}, \qquad (i,t)' \quad \text{(out-node)},$$

and add an internal arc

$$(i,t)'' \to (i,t)'.$$

All arcs that originally entered $(i,t)$, now enter $(i,t)''$, traverse the internal arc $(i,t)'' \to (i,t)'$; and all arcs that originally left $(i,t)$ now emanate from $(i,t)'$. Thus, at each time $t = 1, \ldots, \bar{T} - 1$, a transport entering any vertex $i$ is forced over $(i,t)'' \to (i,t)'$ before it can move on to the next time layer $t + 1$. Let $\bar{\mathcal{G}}$ denote the resulting augmented graph with $(\bar{T} + 1)$ layers of $\mathcal{G}$, with a total of $2K\bar{T} (= K + 2K(\bar{T} - 1) + K)$ vertices due to node-splitting, and with $K(\bar{T} - 1)$ internal arcs added to the arcs in $\mathcal{G}_0^{\bar{T}}$.

Let $\bar{\pi}$ stack all arc transport variables on $\bar{\mathcal{G}}$. It can be verified that all constraints in **P1** can now be written as $\bar{A}\bar{\pi} \leq \bar{\mathbf{b}}$, $-\bar{A}\bar{\pi} \leq -\bar{\mathbf{b}}$, $I\bar{\pi} \leq \mathbf{1}$, and $-I\bar{\pi} \leq 0$, where $\bar{\mathbf{b}} \in \{0, \pm 1\}^{2K\bar{T}}$ is such that it is $-\mu_i$ for each node $(i, 0)$ in layer 0, $\nu_j$ for each node $(j, \bar{T})$ in the last layer $\bar{T}$, and 0 on every intermediate split node. Define

$$\hat{A} := \begin{bmatrix} \bar{A} \\ -\bar{A} \\ I \\ -I \end{bmatrix}, \qquad \hat{\mathbf{b}} := \begin{bmatrix} \bar{\mathbf{b}} \\ -\bar{\mathbf{b}} \\ \mathbf{1} \\ \mathbf{0} \end{bmatrix}.$$

Since incidence matrices of directed graphs are totally unimodular (TU), $\bar{A}$ is TU; see Chapter 19 in (Schrijver, 1986). Consequently, the overall constraint matrix $\hat{A}$ is TU, because TU is preserved under row sign changes and appending rows of $\pm I$. Since $\hat{A}$ is TU, the polyhedron $\{\bar{\pi} : \hat{A}\bar{\pi} \leq \hat{\mathbf{b}}\}$ has integral extreme points. By the TU theorem (Theorem 19.1 and Corollary 19.1a in (Schrijver, 1986)), solving the LP corresponding to **P1** on the augmented $\bar{\mathcal{G}}$ admits an optimal basic solution that is integral. From standard network flow arguments (Ford & Fulkerson, 1962; Ahuja et al., 1993), any feasible flow in the augmented network yields an equivalent feasible flow in the original time-expanded network $\mathcal{G}_0^{\bar{T}}$ that respects the node-capacity constraints, and conversely. Thus, $\{\Pi_t^*\}_{t=1}^{\bar{T}}$ can be chosen as an optimal basic solution and is therefore integral, and (i) follows. The rest of the lemma follows from the standard arguments in linear programming; see also Section 6 on precise complexity arguments. □

## C. Proof of Theorem 3.4

*Proof.* We first show (i). Robots may collide in the following scenarios: (a) at two intersecting edges $i \to m$ and $j \to k$, for distinct $i, j, k, m$; or, (b) a robot traversing $i \to j$ may collide with stationary robots at nearby vertices; or, (c) when two robots travel to the same destination; or, (d) at a bidirectional edge $i \leftrightarrow j$; or, (e) in a flow cycle $i_1 \to i_2 \to \cdots \to i_k \to i_1$, for $k \geq 3$. Clearly, (a) and (b) are ruled out because of Assumption 3.1(ii), and (c) is ruled out by the vertex-capacity constraint. To show that (d) does not appear in the transport, we proceed as follows. Suppose, on the contrary, that there exists a min-cost transport $\{\Pi_t\}_{t=1}^{\bar{T}}$, such that at time $t$ and two distinct vertices $i \neq j$, we have $\pi_{ij,t} = 1$ and $\pi_{ji,t} = 1$, i.e., two robots simultaneously traverse the edges $i \to j$ and $j \to i$. Since self-loops $i \to i$ and $j \to j$ are feasible by Assumption 3.1(i), define an alternate plan $\{\tilde{\Pi}_t\}_{t=1}^{\bar{T}}$ that is exactly the same as the optimal $\{\Pi_t\}_{t=1}^{\bar{T}}$, except for these swaps, which are replaced by waiting moves, i.e.,

$$\tilde{\pi}_{ii,t} = 1, \quad \tilde{\pi}_{jj,t} = 1, \quad \tilde{\pi}_{ij,t} = 0, \quad \tilde{\pi}_{ji,t} = 0.$$

This modification preserves feasibility, since the row and column sums of the $t$-th slice $\Pi_t$ remain unchanged and hence the distributions $\mathbf{q}_{t-1}$ and $\mathbf{q}_t$ are identical for both $\{\Pi_t\}_{t=1}^{\bar{T}}$ and $\{\tilde{\Pi}_t\}_{t=1}^{\bar{T}}$. By Assumption 3.1(v),

$$c_{ii,t} + c_{jj,t} < c_{ij,t} + c_{ji,t},$$

and therefore

$$\langle \tilde{\Pi}_t, C_t \rangle < \langle \Pi_t, C_t \rangle,$$

contradicting the optimality of $\{\Pi_t\}_{t=1}^{\bar{T}}$. With a similar argument, a $k$-cycle leaves the configuration unchanged (because the robots are anonymous) at the price of $k$ moves and is therefore also suboptimal. Hence, no min-cost solution contains a collision and (i) follows; (ii) follows consequently, and (iii) is guaranteed by the terminal feasibility of **P1**. □

We next provide some useful remarks. Note that the claims in Theorem 3.4 are on the optimal (min-cost) solution obtained from **P1**. With the help of Lemma 3.3, integral optimality is guaranteed in polynomial time. In contrast, an integer program implementation of **P1** is intractable in general, forcing early termination within some optimality gap. The resulting suboptimal solution does not necessarily avoid head-on or cycle collisions (as it is not minimum cost), which further requires adding these as hard constraints, thereby breaking TU. That is, enforcing collision avoidance explicitly destroys the structure that could have provided it implicitly.

## D. Proof of Lemma 3.6

*Proof.* That **P1** results in integral transports is already established in Lemma 3.3. Let $\Pi_t' \in \{0, 1\}^{K \times K}$, $t = 1, \ldots, \bar{T}$, be an optimal solution of **P1** obtained with the cost structure $C_t$ described in Assumption 3.5, i.e., $\{\Pi_t'\}_{t=1}^{\bar{T}}$ minimizes the

transport cost $\sum_t \langle \Pi'_t, C_t \rangle$, over the horizon $t = 0, 1, \ldots, \bar{T}$, and let $T' \leq \bar{T}$ be its makespan, i.e., there is no motion after $T'$. Let $T^*$ denote the minimum makespan over all feasible transports, and fix any feasible integral transport $\{\Pi^*_t\}_{t=1}^{\bar{T}}$, whose makespan is $T^*$ (extended to the horizon $\bar{T}$ by waits at targets). Clearly, we have that $T' \geq T^*$ and we need to show that $T' = T^*$.

Suppose, on the contrary, that $T' > T^*$. Since $T'$ is the makespan of $\{\Pi'_t\}_t$, there exists at least one non-wait transport at time $T'$, i.e., $\pi'_{ij,T'} = 1$, for some $i \neq j$, with cost

$$c_{ij,T'} = B^{T'} \tilde{c}_{ij} \geq B^{T'} \tilde{c}_{\min},$$

from Assumption 3.5, and therefore the min-cost transport $\Pi'_t$ satisfies:

$$\sum_{t=1}^{\bar{T}} \langle \Pi'_t, C_t \rangle \geq \langle \Pi'_{T'}, C_{T'} \rangle \geq c_{ij,T'} \geq B^{T'} \tilde{c}_{\min} \geq B^{T^*+1} \tilde{c}_{\min},$$

since $T' \geq T^* + 1$. Similarly, $\{\Pi^*_t\}_t$ has no non-wait motion after time $T^*$, so its cost is supported only on times $t \leq T^*$, and using $\pi^*_{ij,t} \leq 1$, we have

$$\sum_{t=1}^{\bar{T}} \langle \Pi^*_t, C_t \rangle = \sum_{t=1}^{T^*} \langle \Pi^*_t, C_t \rangle \leq \sum_{t=1}^{T^*} \sum_{i \to j \in \mathcal{E}} c_{ij,t} = \sum_{t=1}^{T^*} B^t \sum_{i \to j \in \mathcal{E}} \tilde{c}_{ij}.$$

By Assumption 3.5, it follows that

$$B^{T^*+1} \tilde{c}_{\min} > \frac{B^{T^*+1}}{B-1} \sum_{i \to j \in \mathcal{E}} \tilde{c}_{ij} \geq \sum_{t=1}^{T^*} B^t \sum_{i \to j \in \mathcal{E}} \tilde{c}_{ij}.$$

Hence, $\sum_{t=1}^{\bar{T}} \langle \Pi'_t, C_t \rangle > \sum_{t=1}^{\bar{T}} \langle \Pi^*_t, C_t \rangle$, contradicting the optimality of $\{\Pi'_t\}_t$ and we conclude that $T' = T^*$. $\qquad\square$

## E. Proof of Lemma 4.1

*Proof.* Since $\mathbf{G}$ is Markovian, it admits the following factorization:

$$\mathbf{G}_{i_0,\ldots,i_T} = [\mathbf{g}_0]_{i_0} \prod_{t=1}^{T} \frac{[\mathbf{G}_t]_{i_{t-1},i_t}}{[\mathbf{g}_{t-1}]_{i_{t-1}}}.$$

Recalling the definition of KL divergence, write

$$\mathrm{KL}(\mathbf{P}\|\mathbf{G}) = \sum_{i_0,\ldots,i_T} \mathbf{P}_{i_0,\ldots,i_T} \log \frac{\mathbf{P}_{i_0,\ldots,i_T}}{\mathbf{G}_{i_0,\ldots,i_T}}.$$

Substituting the Markov factorizations in (3) yields

$$\mathrm{KL}(\mathbf{P}\|\mathbf{G}) = \sum_{i_0,\ldots,i_T} \mathbf{P}_{i_0,\ldots,i_T} \left[ \log \frac{\frac{1}{N}[\mathbf{q}_0]_{i_0}}{[\mathbf{g}_0]_{i_0}} + \sum_{t=1}^{T} \left( \log \frac{\frac{1}{N}[\Pi_t]_{i_{t-1},i_t}}{[\mathbf{G}_t]_{i_{t-1},i_t}} - \log \frac{\frac{1}{N}[\mathbf{q}_{t-1}]_{i_{t-1}}}{[\mathbf{g}_{t-1}]_{i_{t-1}}} \right) \right].$$

Distributing the sums and marginalizing $\mathbf{P}$ yields (4) and the proof follows. $\qquad\square$

## F. Proof of Lemma 4.2

*Proof.* Recall that $\mathbf{q}_t$, for any $t = 0, 1, \ldots, T$, is a $\{0,1\}$ vector with exactly $N$ ones. Therefore,

$$\mathrm{KL}(\tfrac{1}{N}\mathbf{q}_t \| \mathbf{g}_t) = \sum_{i=1}^{K} \tfrac{1}{N}[\mathbf{q}_t]_i \log \frac{\frac{1}{N}[\mathbf{q}_t]_i}{[\mathbf{g}_t]_i} = \sum_{i \in \mathcal{V}_t} \frac{1}{N} \log \frac{\frac{1}{N}}{[\mathbf{g}_t]_i} := \kappa_t,$$

for all $t$, since $\mathbf{g}_t$ are given fixed references. From (4), we have

$$\mathrm{KL}(\mathbf{P}\|\mathbf{G}) = \sum_{t=1}^{T} \mathrm{KL}(\tfrac{1}{N}\Pi_t\|\mathbf{G}_t) + \kappa,$$

where $\kappa$ encodes all $\kappa_t$'s and is independent of the transport variables. Fixing $t$ and expanding the KL term, we get:

$$
\begin{aligned}
\mathrm{KL}(\tfrac{1}{N}\Pi_t\|\mathbf{G}_t) &= \sum_{i,j} \tfrac{1}{N}\pi_{ij,t} \log \frac{\tfrac{1}{N}\pi_{ij,t}}{\mathbf{G}_{ij,t}} \\
&= \sum_{i,j} \tfrac{1}{N}\pi_{ij,t} \log \tfrac{1}{N}\pi_{ij,t} - \sum_{i,j} \tfrac{1}{N}\pi_{ij,t} \log \frac{g_{ij,t}}{z_t} \\
&= \sum_{i,j} \tfrac{1}{N}\pi_{ij,t} \log \tfrac{1}{N}\pi_{ij,t} + \sum_{i,j} \tfrac{1}{N}\pi_{ij,t}\frac{c_{ij,t}}{\varepsilon} + \sum_{i,j} \tfrac{1}{N}\pi_{ij,t} \log z_t, \\
&= \tfrac{1}{N}\sum_{i,j} \pi_{ij,t} \log \pi_{ij,t} + \log \tfrac{1}{N} + \tfrac{1}{N}\sum_{i,j} \pi_{ij,t}\frac{c_{ij,t}}{\varepsilon} + \log z_t,
\end{aligned}
$$

which yields the desired result after noting that $\sum_{i,j} \pi_{ij,t} = N$. $\qquad\square$

## G. Sinkhorn-MAPF

Recall that **P2** is an entropic regularization of the LP in **P1**. This regularization enables the use of efficient Sinkhorn-type algorithms, which scale to problem sizes where the original LP may become computationally intractable. We leverage these ideas to design an algorithm for the time-expanded transport formulation. Recall $\mathbf{G}_t$ from Lemma 4.1. It is well known (see e.g., Lemma 2 in (Cuturi, 2013)) that the minimizer of **P2** is obtained at

$$\widetilde{\Pi}_t = \mathrm{diag}(\mathbf{u}_t)\,\mathbf{G}_t\,\mathrm{diag}(\mathbf{v}_t), \qquad t = 1, \ldots, T,$$

for some positive scaling vectors $\mathbf{u}_t, \mathbf{v}_t \in \mathbb{R}_+^K$, where $\mathrm{diag}(\mathbf{u}_t)$ is the diagonal matrix formed by the vector $\mathbf{u}_t$. The Sinkhorn algorithm is an iterative scheme to find the scaling vectors $\mathbf{u}_t, \mathbf{v}_t$ such that $\widetilde{\Pi}_t$, as written above, satisfies the constraint set $\mathcal{F}$. We describe this procedure next. Given the element-wise expansion of the above, i.e.,

$$\widetilde{\pi}_{i,j,t} = \mathbf{u}_{i,t}\,\mathbf{G}_{ij,t}\,\mathbf{v}_{j,t}, \qquad t = 1, \ldots, T,\ i,j = 1, \ldots, K,$$

we note that, for each $i, t$,

$$[\widetilde{\Pi}_t \mathbf{1}]_i = \sum_j \widetilde{\pi}_{ij,t} = \mathbf{u}_{i,t} \sum_j \mathbf{G}_{ij,t}\mathbf{v}_{j,t} = \mathbf{u}_{i,t}\,[\mathbf{G}_t \mathbf{v}_t]_i,$$

and similarly, for each $j, t$,

$$[\widetilde{\Pi}_t^\top \mathbf{1}]_j = \sum_i \widetilde{\pi}_{ij,t} = \mathbf{v}_{j,t} \sum_i \mathbf{u}_{i,t}\,\mathbf{G}_{ij,t} = \mathbf{v}_{j,t}\,[\mathbf{G}_t^\top \mathbf{u}_t]_j.$$

We can now write all constraints in $\mathcal{F}$ in terms of the scaling vectors as follows:

$$
\begin{aligned}
\mathbf{q}_{t-1} &= \widetilde{\Pi}_t \mathbf{1} = \mathbf{u}_t \odot [\mathbf{G}_t \mathbf{v}_t], \\
\mathbf{q}_t &= \widetilde{\Pi}_t^\top \mathbf{1} = \mathbf{v}_t \odot [\mathbf{G}_t^\top \mathbf{u}_t],
\end{aligned}
$$

where $\odot$ denotes the element-wise product. Since $\widetilde{\Pi}_t^\top \mathbf{1} = \mathbf{q}_t = \widetilde{\Pi}_{t+1}\mathbf{1}$, we have the dynamic consistency equation:

$$\mathbf{v}_t \odot [G_t^\top \mathbf{u}_t] = \mathbf{u}_{t+1} \odot [G_{t+1}\mathbf{v}_{t+1}], \qquad t = 1, \ldots, T-1. \tag{5}$$

The boundary conditions are given by

$$\mathbf{q}_0 = \widetilde{\Pi}_1 \mathbf{1} = \mathbf{u}_1 \odot [\mathbf{G}_1 \mathbf{v}_1] = \boldsymbol{\mu}, \tag{6}$$

$$\mathbf{q}_T = \widetilde{\Pi}_T^\top \mathbf{1} = \mathbf{v}_T \odot [\mathbf{G}_T^\top \mathbf{u}_T] = \boldsymbol{\nu}. \tag{7}$$

We now define the Sinkhorn-MAPF algorithm below[3]:

**Initialization:** Set $\mathbf{u}_t^{(0)} = \mathbf{v}_t^{(0)} = \mathbf{1}$ for all $t = 1, \ldots, T$ and then normalize the boundary slices only on the prescribed supports, i.e.,

$$[\mathbf{u}_1^{(0)}]_i \leftarrow \frac{[\boldsymbol{\mu}]_i}{[G_1 \mathbf{v}_1^{(0)}]_i}, \text{ for all } i \in \text{supp}(\boldsymbol{\mu}), \qquad [\mathbf{v}_T^{(0)}]_j \leftarrow \frac{[\boldsymbol{\nu}]_j}{[G_T^\top \mathbf{u}_T^{(0)}]_j}, \text{ for all } j \in \text{supp}(\boldsymbol{\nu});$$

where we adopt the convention throughout that the ratio is set to $0$ if the denominator is zero unless otherwise stated.

**Sinkhorn sweeps:** For $\tau = 0, \ldots, \bar{\tau} - 1$, repeat the following:

1. *Starting marginal projection:* Given $\mathbf{v}_1^{(\tau)}$, update $\mathbf{u}_1^{(\tau+1)}$ to conform with the starting marginal $\boldsymbol{\mu}$, using

$$[\mathbf{u}_1^{(\tau+1)}]_i = \begin{cases} [\boldsymbol{\mu}]_i/[G_1 \mathbf{v}_1^{(\tau)}]_i, & \text{if } [\boldsymbol{\mu}]_i = 1, \\ [\mathbf{u}_1^{(\tau)}]_i, & \text{if } [\boldsymbol{\mu}]_i = 0, \end{cases}$$

and $\mathbf{u}_t^{(\tau+1)} = \mathbf{u}_t^{(\tau)}$, for $t > 1$.

2. *Forward consistency projection:* For each $t = 1, \ldots, T - 1$, enforce the dynamic consistency between $t$ and $t + 1$. First compute the current scaling constants

$$[\mathbf{q}_t^{\text{out}}]_i = [\mathbf{v}_t^{(\tau)}]_i \cdot [G_t^\top \mathbf{u}_t^{(\tau+1)}]_i,$$
$$[\mathbf{q}_t^{\text{in}}]_i = [\mathbf{u}_{t+1}^{(\tau+1)}]_i \cdot [G_{t+1} \mathbf{v}_{t+1}^{(\tau)}]_i,$$

and define a correction factor

$$[\boldsymbol{\gamma}_t]_i = \begin{cases} \sqrt{\dfrac{[\mathbf{q}_t^{\text{in}}]_i}{[\mathbf{q}_t^{\text{out}}]_i}}, & \text{if } [\mathbf{q}_t^{\text{out}}]_i > 0, \\ 1, & \text{otw.} \end{cases}$$

to update

$$[\mathbf{v}_t^{(\tau+1)}]_i = [\mathbf{v}_t^{(\tau)}]_i [\boldsymbol{\gamma}_t]_i^\xi,$$
$$[\mathbf{u}_{t+1}^{(\tau+1)}]_i = [\mathbf{u}_{t+1}^{(\tau+1)}]_i [\boldsymbol{\gamma}_t]_i^{-\xi}.$$

The multiplicative factor $[\boldsymbol{\gamma}_t]_i^\xi$ moves $[\mathbf{q}_t^{\text{out}}]_i$ and $[\mathbf{q}_t^{\text{in}}]_i$ closer at each node $i$. If $\xi = 1$ (and no other constraints are active), then the update enforces $[\mathbf{q}_t^{\text{out}}]_i = [\mathbf{q}_t^{\text{in}}]_i$ in one step; for $\xi \in (0, 1)$, it is a damped projection.

3. *Terminal marginal projection:* Given $\mathbf{u}_T^{(\tau+1)}$, update $\mathbf{v}_T^{(\tau+1)}$ to conform with the terminal marginal $\boldsymbol{\nu}$, using

$$[\mathbf{v}_T^{(\tau+1)}]_j = \begin{cases} [\boldsymbol{\nu}]_j/[G_T^\top \mathbf{u}_T^{(\tau+1)}]_j, & \text{if } [\boldsymbol{\nu}]_j = 1, \\ [\mathbf{v}_T^{(\tau)}]_j, & \text{if } [\boldsymbol{\nu}]_j = 0. \end{cases}$$

For $t < T$, we already have $\mathbf{v}_t^{(\tau+1)}$ from the consistency update above.

4. *Backward consistency projection:* For each $t = T - 1, \ldots, 1$, enforce the dynamic consistency between $t + 1$ and $t$, i.e., in reverse time. Compute

$$[\bar{\mathbf{q}}_t^{\text{out}}]_i = [\mathbf{u}_{t+1}^{(\tau+1)}]_i \cdot [G_{t+1} \mathbf{v}_{t+1}^{(\tau+1)}]_i,$$
$$[\bar{\mathbf{q}}_t^{\text{in}}]_i = [\mathbf{v}_t^{(\tau+1)}]_i \cdot [G_t^\top \mathbf{u}_t^{(\tau+1)}]_i,$$

---

[3]It is not uncommon to implement the resulting equations in the log domain to account for the numerical instabilities when $\varepsilon \to 0$, see e.g., (Schmitzer, 2019).

and

$$[\bar{\boldsymbol{\gamma}}_t]_i = \begin{cases} \sqrt{\dfrac{[\bar{\mathbf{q}}_t^{\text{in}}]_i}{[\bar{\mathbf{q}}_t^{\text{out}}]_i}}, & [\bar{\mathbf{q}}_t^{\text{out}}]_i > 0, \\ 1, & \text{otw.} \end{cases}$$

with updates

$$[\mathbf{u}_{t+1}^{(\tau+1)}]_i = [\mathbf{u}_{t+1}^{(\tau+1)}]_i\,[\bar{\boldsymbol{\gamma}}_t]_i^{\xi},$$
$$[\mathbf{v}_t^{(\tau+1)}]_i = [\mathbf{v}_t^{(\tau+1)}]_i\,[\bar{\boldsymbol{\gamma}}_t]_i^{-\xi}.$$

As in the forward sweep, the multiplicative factor $[\bar{\boldsymbol{\gamma}}_t]_i^{\xi}$ reduces the discrepancy between $[\mathbf{q}_t^{\text{out}}]_i$ and $[\mathbf{q}_t^{\text{in}}]_i$ at each node $i$, but now propagates corrections backward in time.

**Final regularized transports:** The algorithm is terminated at $\tau = \bar{\tau} - 1$, resulting in

$$\widetilde{\Pi}_t = \text{diag}(\mathbf{u}_t^{(\bar{\tau})})\,G_t\,\text{diag}(\mathbf{v}_t^{(\bar{\tau})}), \qquad t = 1, \ldots, T,$$

## H. Detailed Experiments

In this section, we provide a detailed set of experiments. We consider square grids of size $K = W^2$ with $N$ robots, $M$ targets, and $O$ obstacles, such that $N = M$. While the proposed framework applies to arbitrary graphs, grids are chosen for ease of visualization and reproducibility. The time horizons are chosen such that the corresponding LPs are feasible; smaller values of $T$ are only chosen for convenience as it is easier to display the corresponding trajectories. Unless stated otherwise, the cost structure is such that waiting cost at targets is $c_{ii} = 0, i \in \mathcal{M}$, waiting at non-target is $c_{jj} = 0.5, j \notin \mathcal{M}$, while the move cost is $c_{i \neq j} = 1$, for all $i \to j \in \mathcal{E}$.

### H.1. Illustrative Experiments

We first provide some basic experiments to demonstrate the main ideas. The experiment setup is described in the caption of each figure and the corresponding LPs are solved using the standard linear programming suite in the HiGHS solver from the SciPy linprog library; all variables are continuous by default. Figs. 9, 10, and 11 solve **P1** over a given time horizon $T$; we note that arbitrarily placed obstacles alter the connectivity and diameter of the underlying graph, and it no longer remains a regular grid. Figs. 12-15 elaborate the min-cost versus minimum makespan nature of the transports returned by **P1**. Figs. 16, 17, and 18 solve **P2** with the Sinkhorn iterations of Appendix G for the following $(\varepsilon, \bar{\tau})$ pairs, where $\bar{\tau}$ is the total number of Sinkhorn iterations: $\{(0.2, 50), (0.5, 10), (50, 5)\}$; then prune the resulting graph and project on the integral **P3** LP. As known for Sinkhorn iterations, **P2** may require a larger $\bar{\tau}$ as $\varepsilon$ decreases; however, a useful shadow transport is typically obtained within a small number of iterations.

In the following subsections (Sections H.2–H.5), we conduct a large-scale study of the proposed approaches. We ran over 460 experiments on graphs ranging from $K = 2,500$ to $K = 22,500$ vertices, i.e., from 369K to 3.4M LP variables, using the Gurobi LP solver with continuous variables; all **P1** and **P3** solutions are verified integral as guaranteed by the TU of the underlying LP.

### H.2. Scaling Study

We evaluate the runtime scaling of **P1** optimal and the scalable **P2+P3** pipeline on a 2022 MacBook; all LPs use continuous variables. We consider square grids of size $K = W^2$ with no obstacles, at 5% robot density ($N = 0.05K$), over a horizon $T = 30$. The cost structure follows the convention adopted in Assumption 3.1. The Sinkhorn parameters are $\varepsilon = 0.2$ and $\bar{\tau} = 150$ sweeps. For each grid size, we generate 14–25 random instances and report averaged results. On grid graphs, each vertex has at most 5 neighbors (4 cardinal directions plus a self-loop), so the number of variables in **P1** is $|\mathcal{E}|T \approx 5KT$. For example, at $K = 22,500$ with $T = 30$: $|\mathcal{E}|T = 3,357,000$.

Table 2 summarizes the results across 8 grid sizes, totaling 162 runs. For each grid, we report the number of **P1** variables ($= |\mathcal{E}|T$), the average **P1** solve time to optimality, the average **P2+P3** solve times (Sinkhorn + **P3**), the resulting

*Table 2.* Scaling of **P1** and **P2+P3** across grid sizes at $5\%$ robot density ($N = 0.05K$), $T = 30$, $\varepsilon = 0.2$. All times are averaged and are reported in seconds (s). Every solution across all 162 runs is verified integral.

| $K$ | $N$ | Runs | **P1** vars | **P1** optimal (s) | **P2+P3** (s) | Speedup | Gap (%) | Kept (%) |
|-----|-----|------|-------------|--------------------|---------------|---------|---------|----------|
| 2,500 | 125 | 20 | 369K | 15 | 4 | 3.6× | 8.0 | 32 |
| 5,625 | 281 | 25 | 835K | 55 | 11 | 5.0× | 8.7 | 35 |
| 8,100 | 405 | 19 | 1.2M | 103 | 18 | 5.7× | 9.0 | 37 |
| 10,000 | 500 | 24 | 1.5M | 132 | 26 | 5.1× | 5.9 | 41 |
| 13,225 | 661 | 14 | 2.0M | 193 | 33 | 5.8× | 8.1 | 39 |
| 15,625 | 781 | 23 | 2.3M | 257 | 48 | 5.3× | 5.9 | 43 |
| 19,600 | 980 | 18 | 2.9M | 364 | 62 | 5.8× | 7.2 | 42 |
| 22,500 | 1,125 | 19 | 3.4M | 478 | 67 | 7.1× | 6.5 | 40 |

speedup, cost gap relative to the **P1** optimal, and the percentage of edges retained from the shadow transport (which also provides the average number of **P3** variables, kept after pruning).

Fig. 4 (in Section 7, left panel) plots the **P1** and **P2+P3** solve times against the number of grid vertices $K$, with power-law fits of the form $aK^p + b$ over all 162 individual runs. The fitted models are

$$\textbf{P1:} \quad \text{solve time} = 2.26{\times}10^{-5}\,K^{1.68} + 10.2 \quad (R^2 = 0.96),$$

$$\textbf{P2+P3:} \quad \text{solve time} = 7.40{\times}10^{-4}\,K^{1.15} - 2.97 \quad (R^2 = 0.74).$$

We note that **P1** solve time to optimality grows sub-quadratically in $K$, while the **P2+P3** solve time scales almost linearly. The speedup, shown in Fig. 4 (right panel), grows with problem size from $3.6\times$ at $K = 2,500$ to $7.1\times$ at $K = 22,500$, while the cost gap remains consistently below $10\%$ (median $6.4\%$). Since **P1** already provides the exact optimum, the **P2+P3** pipeline offers a practical speed-quality tradeoff: a 5–7× speedup at under $10\%$ cost degradation.

Fig. 5 examines the cost-gap-versus-speedup tradeoff from two perspectives. The left panel plots the cost gap against the speedup for each of the 162 individual runs, colored by the number of vertices $K$. The cluster structure confirms that larger instances achieve higher speedups at comparable or lower cost gaps, i.e., the shadow-based pruning becomes more effective as $K$ grows. The right panel plots the average cost gap and speedup jointly against $K$ on a dual axis. The cost gap remains stable between 5–9% across all grid sizes, while the speedup increases steadily from $3.6\times$ to $7.1\times$.

Fig. 6 provides a complementary view of the pipeline. The left panel decomposes the average solve time into **P1** (gray), Sinkhorn (blue), and **P3** LP (red) components. The connected dots trace the scaling shape: the **P1** curve grows subquadratically while the **P2+P3** curve remains nearly flat. The italic percentages above each stacked bar indicate the Sinkhorn share of the **P2+P3** time, which decreases from $63\%$ at $K = 2,500$ to $34\%$ at $K = 22,500$; at large scale, the LP solve dominates. The right panel shows the variable reduction achieved by shadow-based pruning: **P3** consistently operates on 32–43% of the **P1** variables.

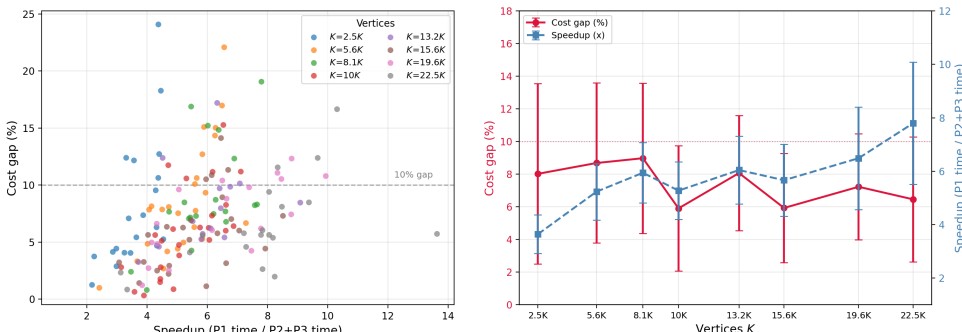

*Figure 5.* Cost gap and speedup tradeoff across 162 scaling runs at $5\%$ robot density, $T = 30$. (Left) Cost gap (%) versus speedup for each individual run, colored by the number of vertices $K$; the dashed line marks the $10\%$ gap threshold. (Right) Average cost gap (%, left axis, red) and speedup (right axis, blue) versus $K$, with $\pm 1$ standard deviation error bars; the dotted line marks the $10\%$ gap threshold.

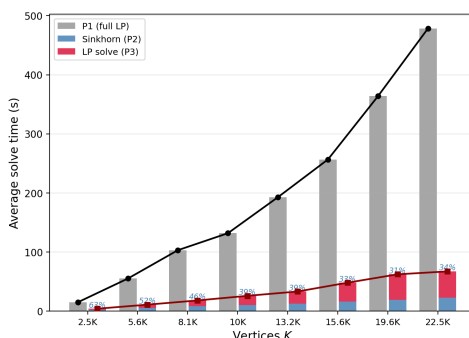 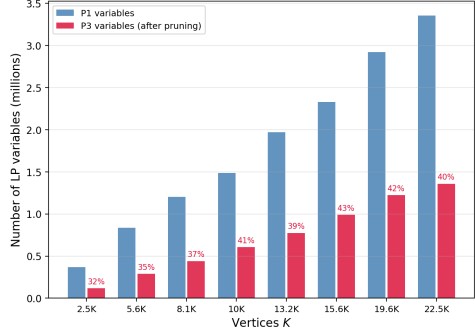

*Figure 6.* Pipeline decomposition across 8 grid sizes at 5% robot density, $T = 30$; all values are averages over 14–25 independent instances. (Left) Solve time of **P1** (gray bars) versus the Sinkhorn (**P2**, blue) and LP solve (**P3**, red) components of the pipeline; connected dots trace the scaling shape of each; italic percentages show the Sinkhorn share of the **P2+P3** time. (Right) Number of LP variables in **P1** (blue) versus **P3** after pruning (red); percentages indicate the fraction of **P1** variables retained.

### H.3. Sinkhorn and Pruning Parameter Sensitivity

In this section, we study the sensitivity of the scalable **P2+P3** pipeline to the corresponding parameters $\varepsilon$ and $\lambda$. We sweep $\varepsilon \in \{0.1, 0.2, 0.5, 1.0, 5.0\}$ and $\lambda \in \{0, 0.5, 1.0, 5.0\}$ over 13 independent random instances on $K = 10{,}000$ vertices ($W = H = 100$, $N = 500$, $T = 30$, 1.5M variables in **P1**), for a total of 260 runs (20 parameter combinations per instance). The **P1** baseline solves to optimality on average in 130s. Of the 260 runs, 233 are feasible; the 27 infeasible cases arise at the extremes $\varepsilon = 0.1$ (19 runs across 5 instances) and $\varepsilon = 5.0$ (8 runs across 2 instances). These infeasibilities are artifacts of the fixed pruning threshold ($\approx 40$–$45\%$ of edges retained); relaxing the threshold to retain more edges recovers feasibility in all cases. Every feasible solution is verified integral.

Table 1 (in Section 7) reports the average cost gap (%) relative to the **P1** optimal for each $(\varepsilon, \lambda)$ combination. The parameter $\varepsilon$ is the dominant factor: small $\varepsilon$ ($\leq 0.2$) produces a concentrated shadow close to the **P1** optimum, enabling aggressive pruning with 2–5% gap, while large $\varepsilon$ smooths the shadow and increases the gap to 17–20%. The parameter $\lambda$ has a milder effect, adding roughly 1–6% to the gap depending on $\varepsilon$. A plausible time-quality tradeoff is at $\varepsilon = 0.2$, $\lambda = 0$: a 4.3% gap in 26s ($5.0\times$ speedup over **P1**).

Table 3 reports the Sinkhorn convergence and **P2 + P3** timing per $\varepsilon$ (at $\lambda = 0$), where Sinkhorn iterations are terminated at convergence. The Sinkhorn convergence scales inversely with $\varepsilon$, as expected from entropic regularization: 307 sweeps at $\varepsilon = 0.1$ versus 39 at $\varepsilon = 5.0$. However, the cost gap increases as $\varepsilon$ increases (from $\sim 2\%$ at $\varepsilon = 0.1$ to $\sim 17\%$ at $\varepsilon = 5.0$), reflecting the smoothing effect that makes the shadow less discriminative and the pruned graph less targeted. The total **P2+P3** time ranges from 48s at $\varepsilon = 0.1$ to 22–26s at $\varepsilon \geq 0.2$, all well below the **P1** baseline of 130s. At $\varepsilon = 5.0$, the runtime increase is because of the diffuse shadow that makes pruning ineffective, resulting in more variables retained in **P3**.

*Table 3.* Sinkhorn convergence and **P2+P3** pipeline timing per $\varepsilon$ at $\lambda = 0$, averaged over 13 instances at $K = 10{,}000$, $T = 30$.

| $\varepsilon$ | Sweeps | Sinkhorn (s) | **P2+P3** (s) | Gap (%) | Kept (%) |
|---|---|---|---|---|---|
| 0.1 | 307 | 20 | 48 | 2.3 | 47 |
| 0.2 | 132 | 9 | 26 | 4.3 | 42 |
| 0.5 | 97 | 7 | 22 | 11.1 | 40 |
| 1.0 | 75 | 5 | 23 | 17.3 | 43 |
| 5.0 | 39 | 3 | 25 | 17.1 | 47 |

**Parameter roles.** The parameter $\varepsilon$ controls the sharpness of the shadow transport: as $\varepsilon \to 0$, the Schrödinger bridge concentrates onto minimum-cost geodesic corridors, producing a sharp shadow that enables aggressive pruning at low cost gap; as $\varepsilon$ increases, the shadow becomes diffuse and the pruned graph retains more edges with less discriminative structure. The parameter $\lambda$ controls the bias toward the shadow in the **P3** cost: increasing $\lambda$ penalizes edges with small shadow flow, effectively forcing **P3** to follow the shadow transport more closely. When costs are uniform, $\lambda$ and $\varepsilon$ together provide a beneficial tiebreaker among edges that are otherwise equivalent in cost, favoring those with a darker shadow. The parameter $\delta$ is a small numerical safeguard for the logarithm in the modified **P3** cost and has negligible effect on the solution. Across the experiments reported in this paper, the following is a robust default that works without much

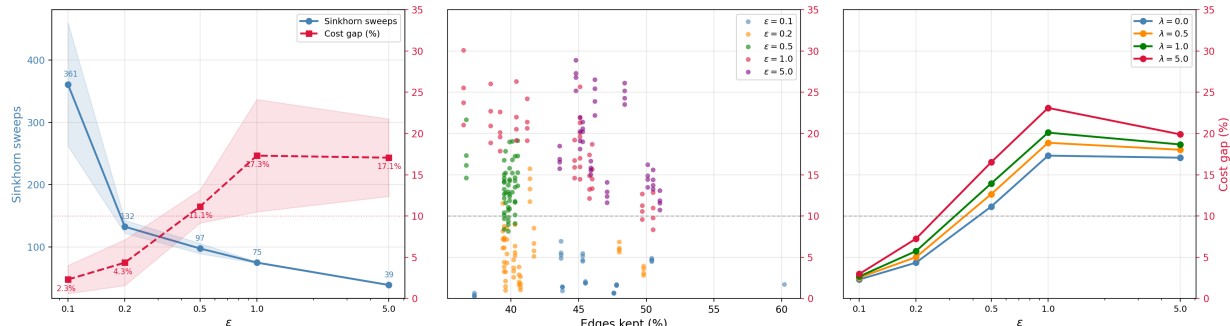

*Figure 7.* Parameter sensitivity across 260 runs (13 instances, 20 combinations each) at $K = 10{,}000$, $T = 30$. (Left) Dual-axis plot at $\lambda = 0$: effective Sinkhorn sweeps (left axis, blue) and cost gap (%, right axis, red) versus $\varepsilon$; shaded bands show $\pm 1$ standard deviation across instances. (Middle) Cost gap versus edges kept (%) for all 233 feasible runs, colored by $\varepsilon$. (Right) Average cost gap versus $\varepsilon$ for each $\lambda$.

fine-tuning: $\delta = 10^{-6}$, $\varepsilon = 0.2$, stop Sinkhorn when the last 20 iterates stabilize, and $\lambda = 0$. For the pruning threshold $\eta$, retaining 40–48% of edges (depending on $\varepsilon$) consistently yields feasible **P3** solutions.

Fig. 7 visualizes the sensitivity structure from three perspectives. The left panel is a dual-axis plot: the left axis shows the effective number of Sinkhorn sweeps (blue) and the right axis shows the cost gap (red), both versus $\varepsilon$ at $\lambda = 0$, with $\pm 1$ standard deviation shaded across the 13 instances. The sweeps decrease inversely with $\varepsilon$, while the gap increases nearly monotonically. The middle panel plots the gap against the fraction of edges retained for all 233 feasible runs, colored by $\varepsilon$, where lower $\varepsilon$ achieves lower gaps at comparable pruning levels. For a fixed $\varepsilon$, the cost variation is also due to varying $\lambda$. The right panel isolates the effect of $\lambda$ by plotting the gap versus $\varepsilon$ for each $\lambda$ value; the curves confirm that $\lambda$ shifts the gap upward by a roughly constant offset, with a mild effect relative to $\varepsilon$.

### H.4. Non-uniform Costs

In this section, we validate the proposed **P1** and the **P2+P3** pipeline under non-uniform costs. We assign each cell a random arrival cost $\sim \mathrm{Uniform}[0.6, 1]$ and wait cost $\sim \mathrm{Uniform}[0.1, 0.5]$ (wait at target = 0), preserving Assumption 3.1. Choosing costs like this reflects e.g., uneven terrains; see Fig. 8 (left) for a candidate scenario. We run 24 instances at $K = 10{,}000$ ($W = H = 100$, $N = 500$ robots, $T = 30$) with $\varepsilon = 0.2$, $\lambda = 0$. The **P1** baseline averages 138s; the **P2+P3** pipeline averages 25s at 5.1% gap and $5.4\times$ speedup. For reference, the uniform-cost baseline (from the sensitivity study) gives 4.3% gap and $5.0\times$ speedup. The gap and speedup under non-uniform costs are comparable, confirming that the pipeline adapts to the cost landscape without degradation. Fig. 8 (middle) shows the per-instance gap and speedup, and Fig. 8 (right) compares the uniform and non-uniform averages. Every solution is verified integral.

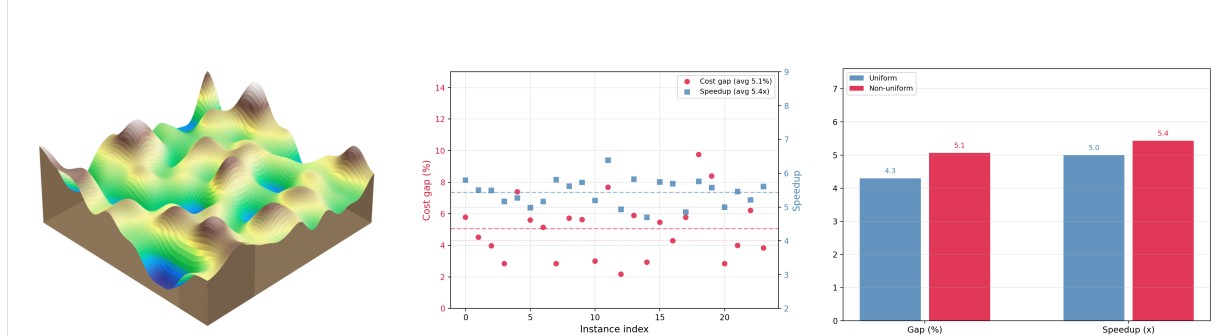

*Figure 8.* Non-uniform cost experiments across 24 instances at $K = 10{,}000$, $T = 30$. (Left) Illustrative terrain on a $100 \times 100$ grid; each cell has a random arrival cost $\sim \mathrm{Uniform}[0.6, 1]$; higher elevation corresponds to higher move cost. (Middle) Cost gap (%, left axis, red) and speedup (right axis, blue) per instance; dashed lines show non-uniform averages (5.1% gap, $5.4\times$ speedup), dotted lines show the uniform-cost reference (4.3% gap, $5.0\times$ speedup). (Right) Uniform versus non-uniform cost comparison, averaged over all instances.

## H.5. Baseline Comparison

In this section, we compare with a related formulation (Ma & Koenig, 2016), which studies TAPF (combined target-assignment and path-finding). In TAPF, agents are partitioned into teams and each team is given the same number of targets as agents. The goal is to jointly assign agents to targets and plan collision-free paths that minimize makespan. The method CBM (Conflict-Based Min-Cost-Flow) proposed therein uses a min-cost max-flow solver on a time-expanded network on the low level (for within-team assignment and routing) and conflict-based search on the high level (for inter-team collision resolution). TAPF with a single team (all agents exchangeable) is the anonymous MAPF problem.

The comparison next provides a useful reference point. We adopt the same experimental setting as (Ma & Koenig, 2016): a $30 \times 30$ grid with $10\%$ randomly blocked cells and 4-neighbor connectivity. Table 4 compares the results. CBM solves up to 50 agents in 5.32s, while the reported ILP-based solver handles 50 agents in 162s with only $4\%$ success rate with a 5-minute timeout; restricting to a single team may improve performance, as the multi-team structure introduces additional complexity. In contrast, the proposed **P1** solves 300 agents (same grid, $6\times$ more agents) in 0.54s on average, and the **P2+P3** pipeline solves in 0.49s at $0.63\%$ gap. Our framework further scales to $K = 22{,}500$ vertices (1,125 agents), where **P2+P3** solves all instances in 67s on average.

*Table 4.* Comparison on $30 \times 30$ grids with $10\%$ obstacles. CBM and ILP results are from Table 1 in (Ma & Koenig, 2016), which reports multi-team TAPF instances. Our results are on single-team anonymous MAPF, averaged over 15 instances.

| Method | Agents | Time (s) | Success |
|---|---|---|---|
| CBM (Ma & Koenig, 2016) | 50 | 5.32 | 100% |
| ILP (Ma & Koenig, 2016) | 50 | 162 | 4% |
| ILP (Ma & Koenig, 2016) | 40 | 153 | 14% |
| **P1** (ours) | 300 | 0.54 | 100% |
| **P2+P3** (ours) | 300 | 0.49 | 100% |

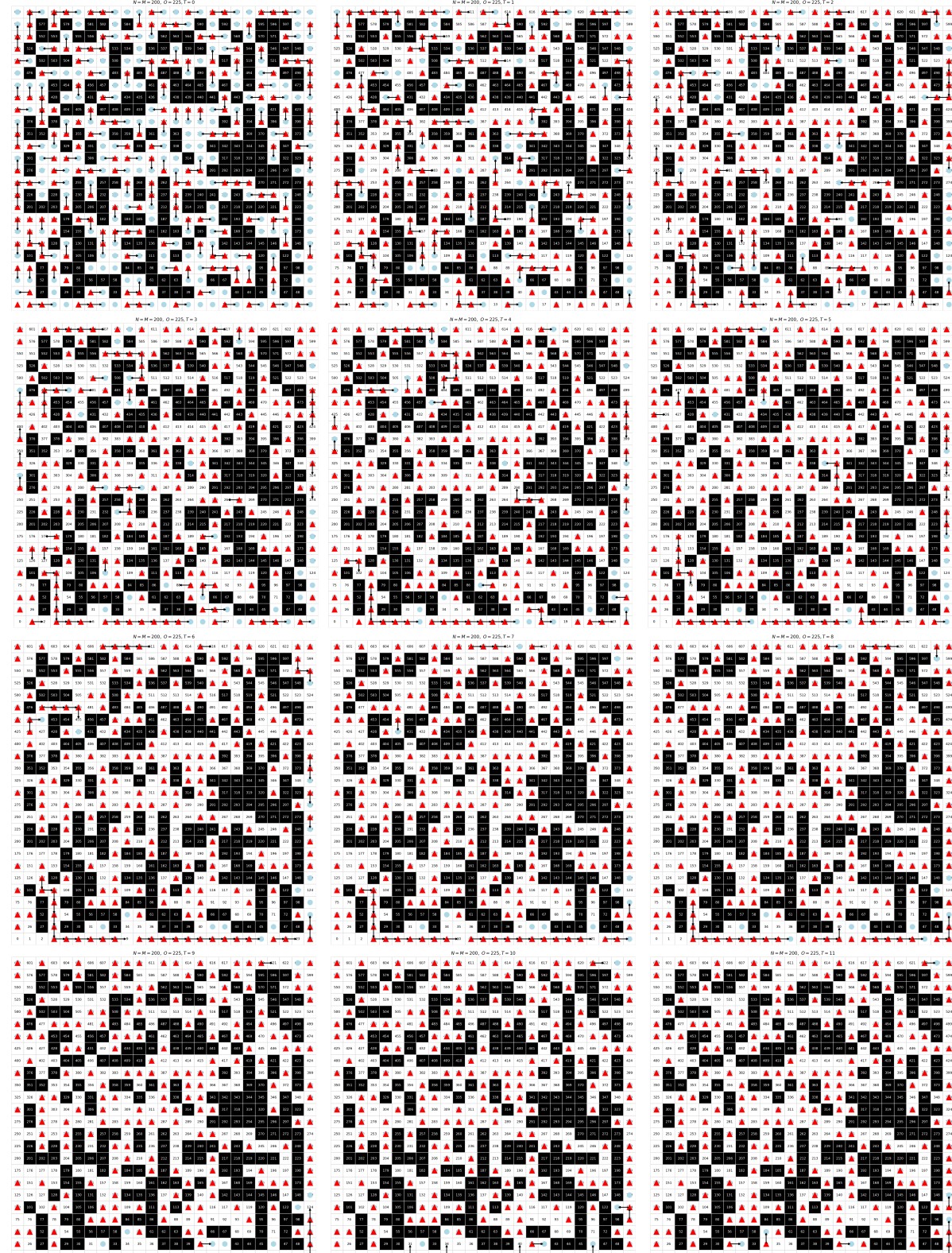

*Figure 9.* A $25 \times 25$ grid with $\{200\}$ robots ▲, $\{200\}$ targets ●, and $\{225\}$ obstacles. Every cell is either a robot, a target, or an obstacle. The robot trajectories come from the min-cost (optimal) transport obtained from **P1** over $T = 12$.

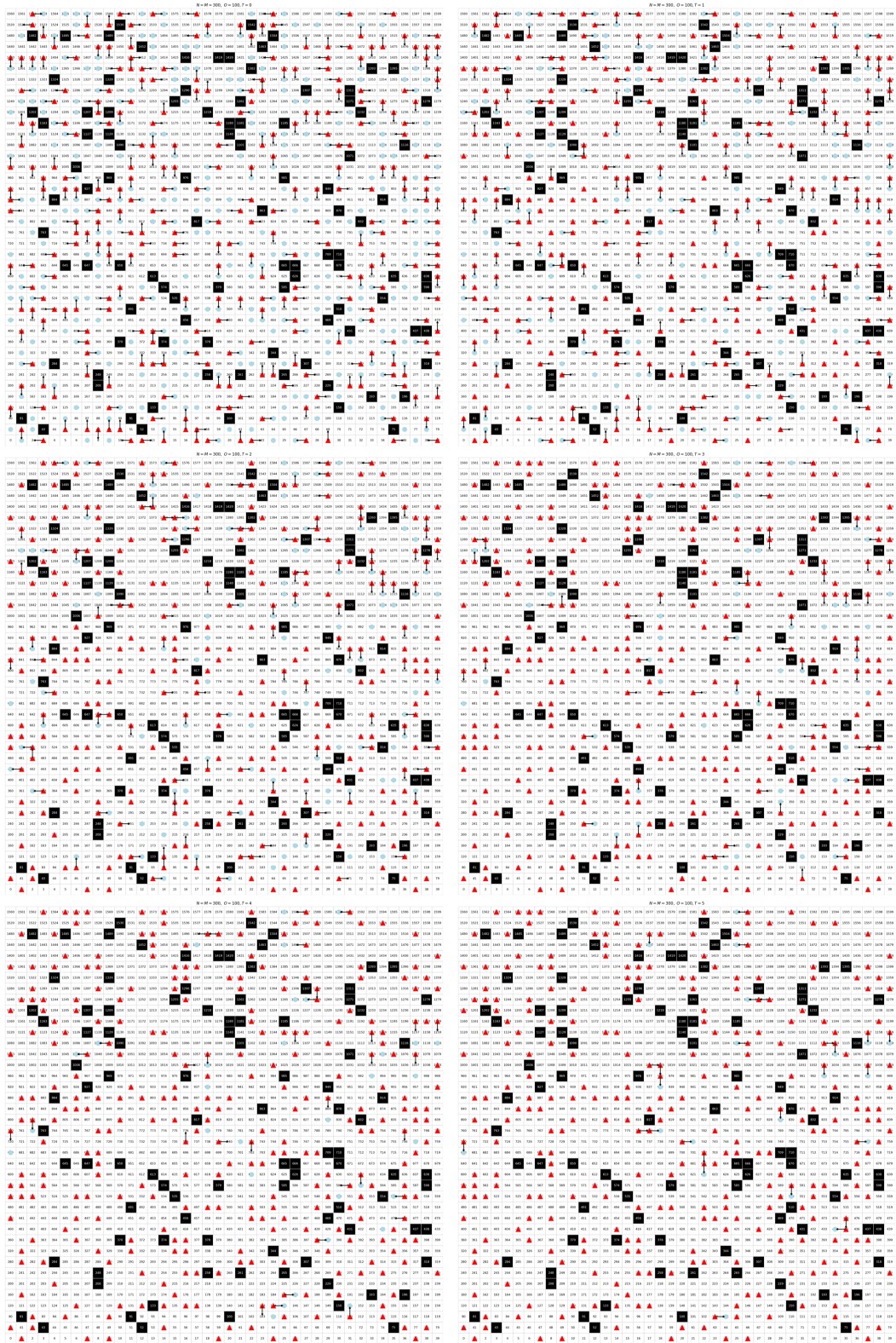

*Figure 10.* A larger $40 \times 40$ grid with 300 robots ▲, 300 targets ●, and 100 obstacles. The robot trajectories come from the min-cost (optimal) transport obtained from **P1** over $T = 6$.

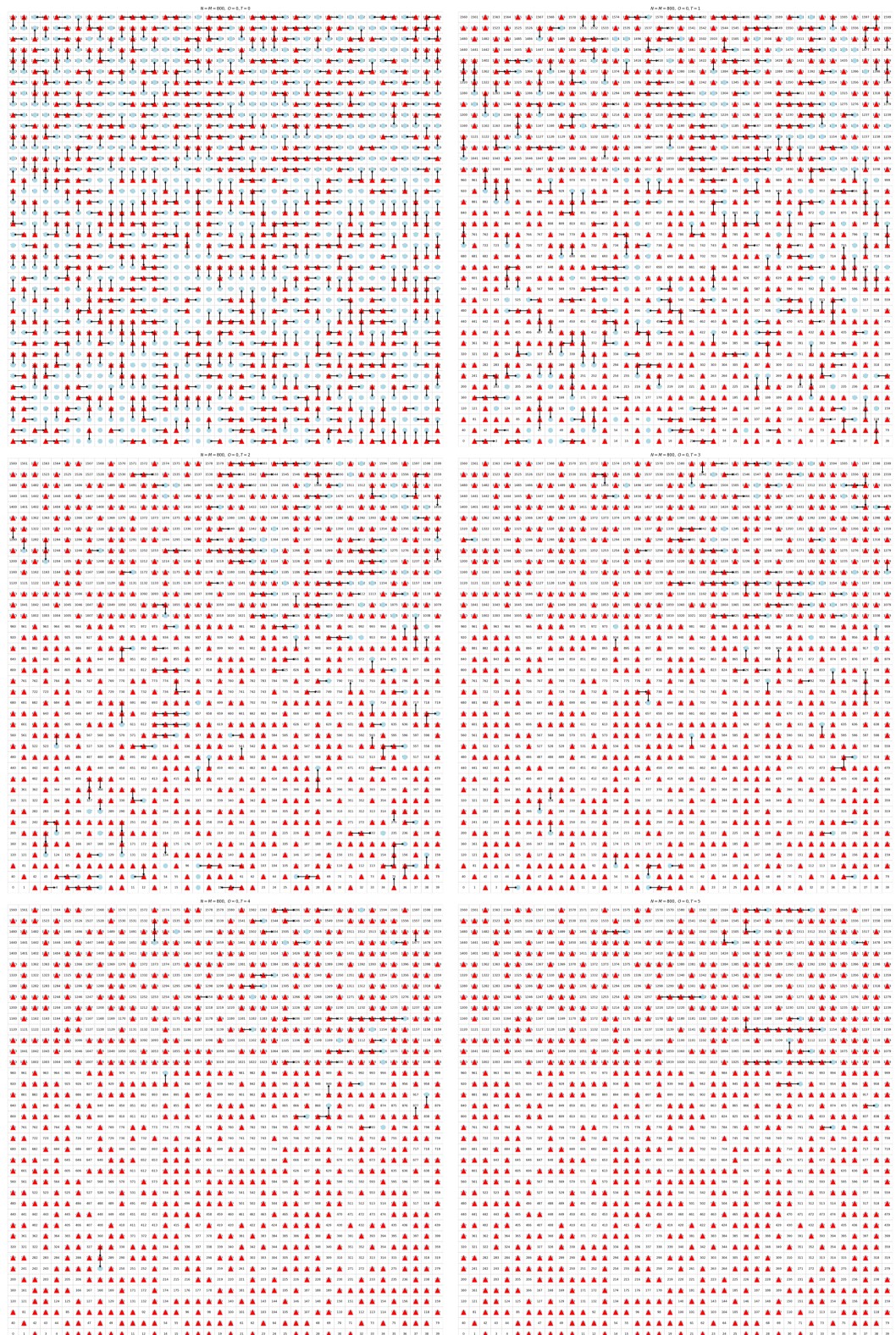

*Figure 11.* A larger $40 \times 40$ grid with 800 robots ▲ and 800 targets ●; every cell is either occupied by a robot or a target. The robot trajectories come from the min-cost (optimal) transport obtained from **P1** over $T = 6$.

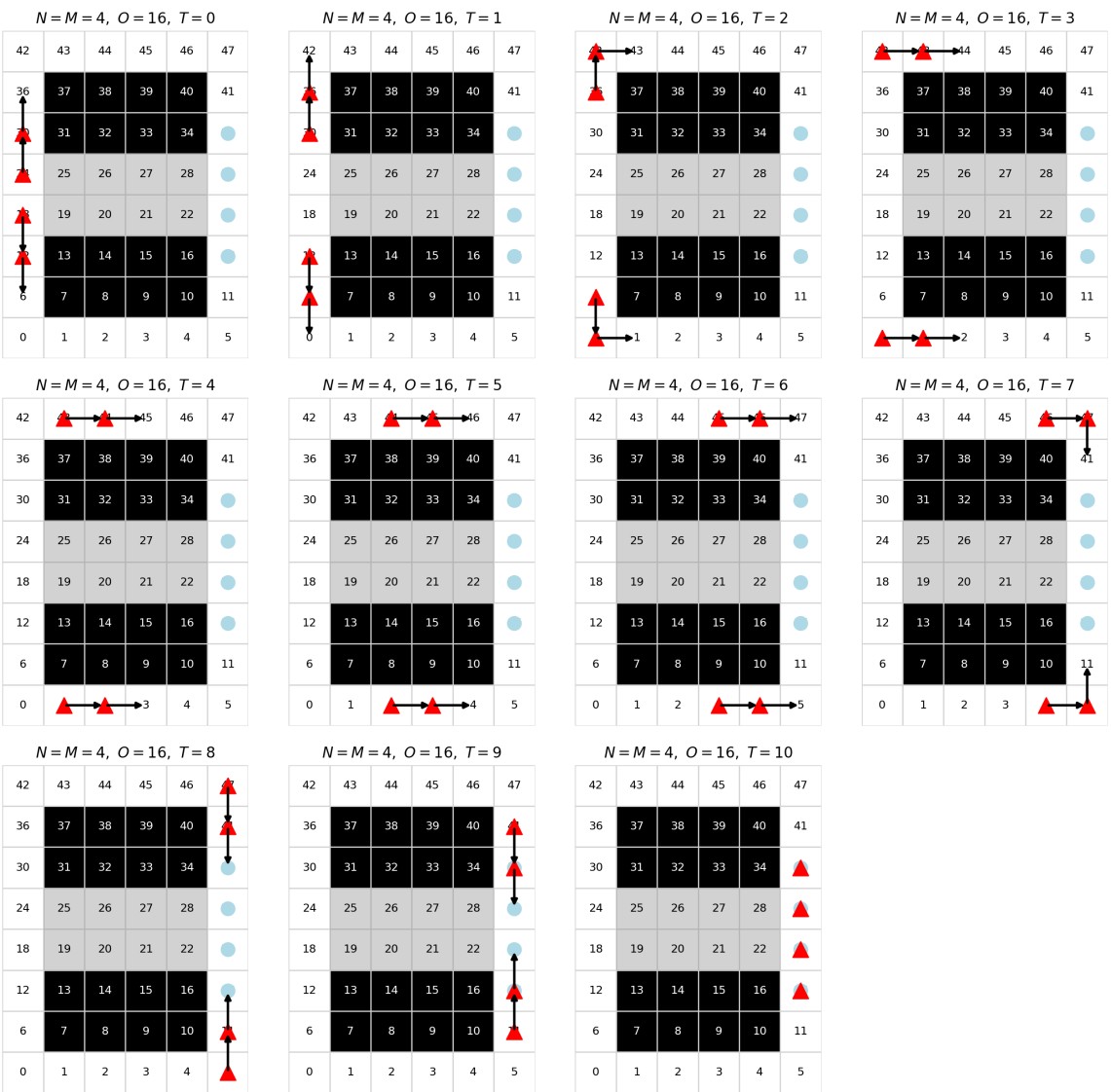

*Figure 12.* Min-cost vs. Min-makespan: A $6 \times 8$ grid with 4 robots ▲, 4 targets ●, and 16 obstacles. Edges in the gray shaded region have cost 10; rest follow our move-wait cost convention.

With $T = 10$, the min-cost transport avoids the high cost interior and takes the robots from the boundary of the grid, with all one-cost moves for a total cost of 40.

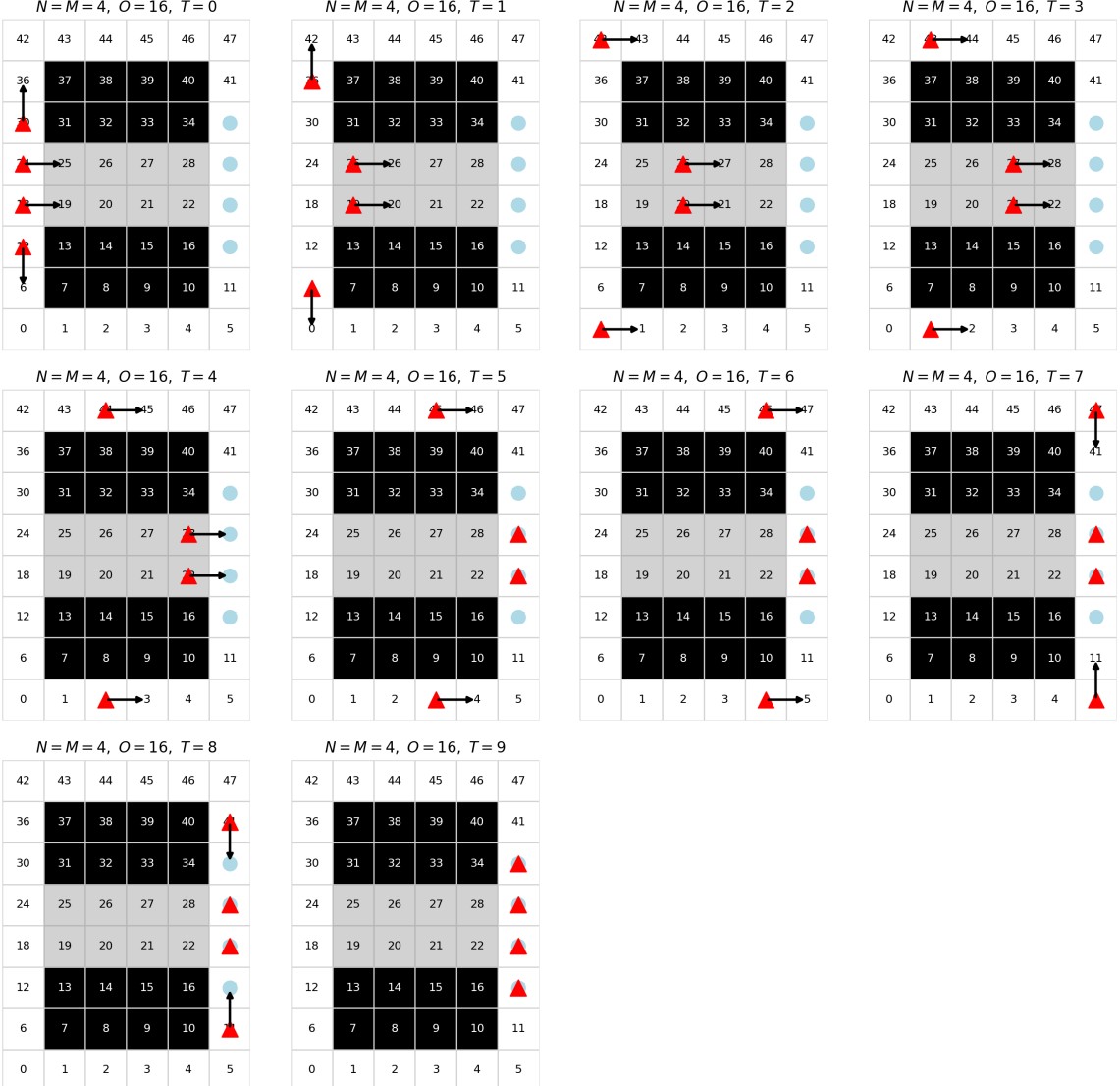

*Figure 13.* Min-cost vs. Min-makespan: A $6 \times 8$ grid with 4 robots ▲, 4 targets ●, and 16 obstacles. Edges in the gray shaded region have cost 10; rest follow our move-wait cost convention.

With $T = 9$, all robots cannot travel on the boundary as that requires 10 moves; the min-cost transport therefore takes two robots from the boundary in 9 steps each, and two from the higher cost interior edges for a total cost of 82.

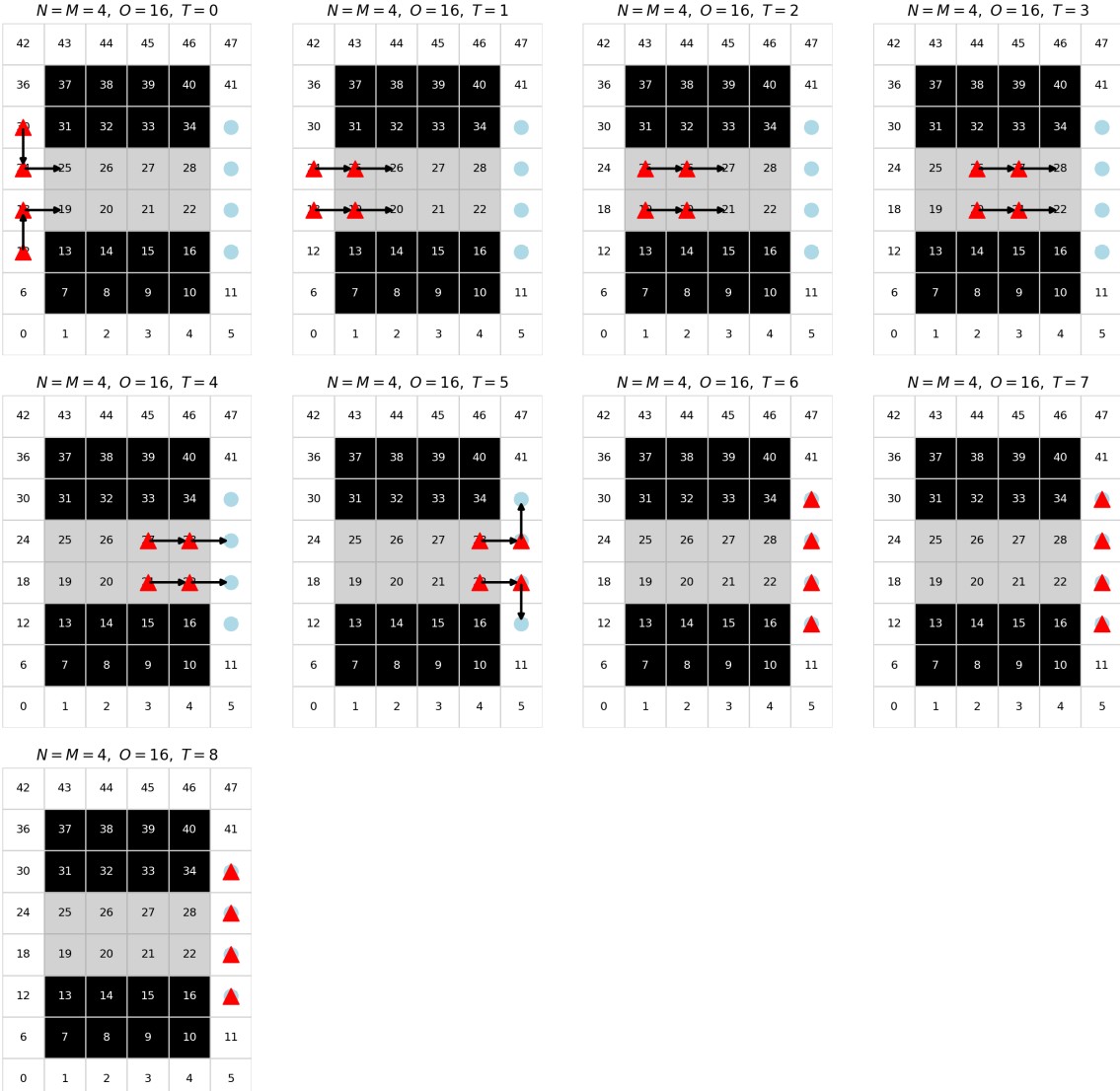

*Figure 14.* Min-cost vs. Min-makespan: A $6 \times 8$ grid with 4 robots ▲, 4 targets ●, and 16 obstacles. Edges in the gray shaded region have cost 10; rest follow our move-wait cost convention.

With $T = 8$, boundary paths are no longer feasible within the given time horizon. All robots must travel through the interior edges to reach the targets for a total cost of 132.

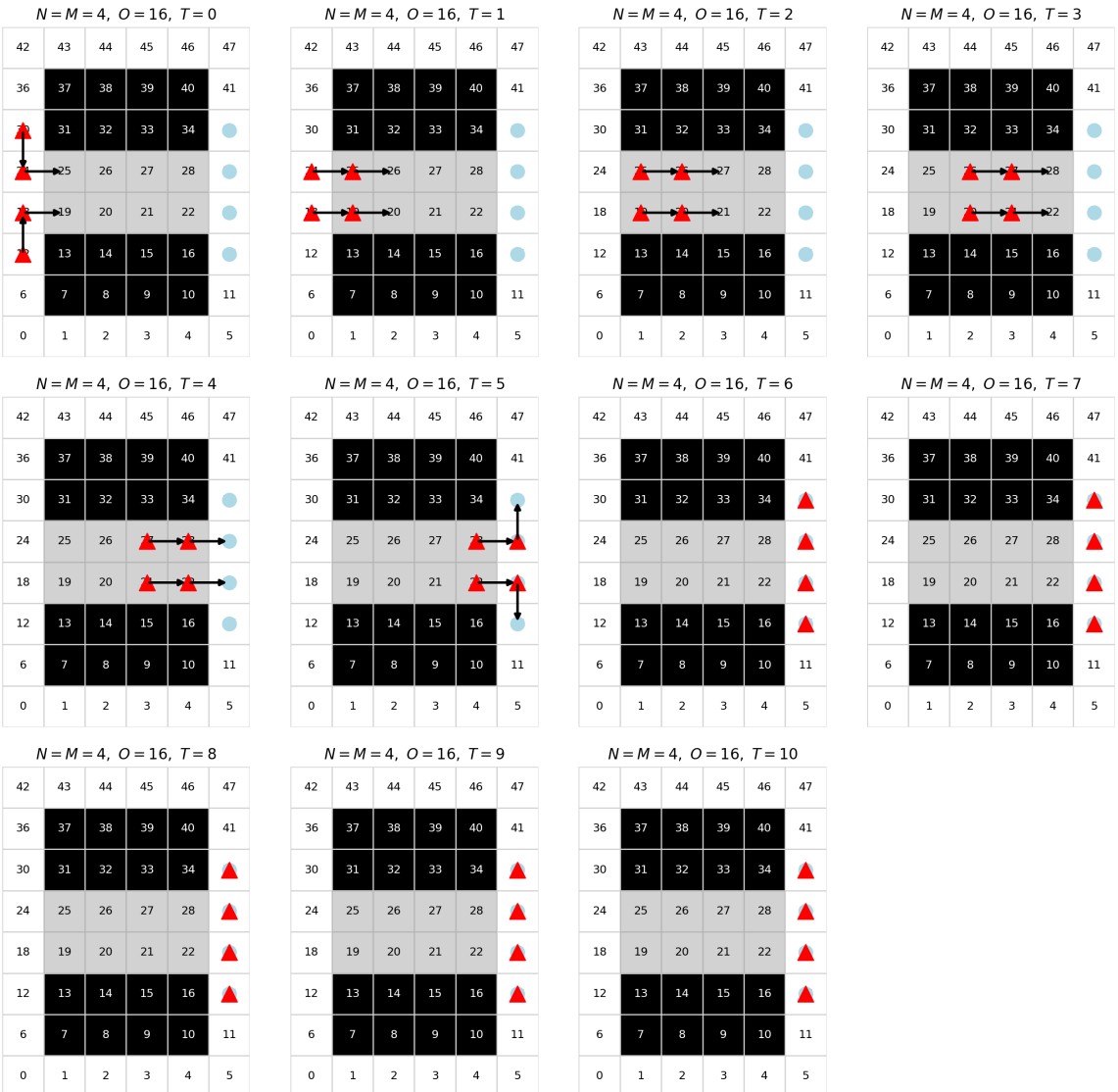

*Figure 15.* Min-cost vs. Min-makespan: A $6 \times 8$ grid with 4 robots ▲, 4 targets ●, and 16 obstacles. Edges in the gray shaded region have cost 10; rest follow our move-wait cost convention.

We choose the cost structure described in Assumption 3.5. **P1** consequently provides the minimum makespan solution that terminates the robot motion in 5 steps, i.e., achieves the minimum makespan, when solved over a longer $T = 10$ horizon. The total cost of this transport increases exponentially, of the order of $10^{13}$. For larger problems, the corresponding solvers may run into numerical instabilities and the search procedure described in Lemma 3.7 may be more tractable to compute the minimum makespan.

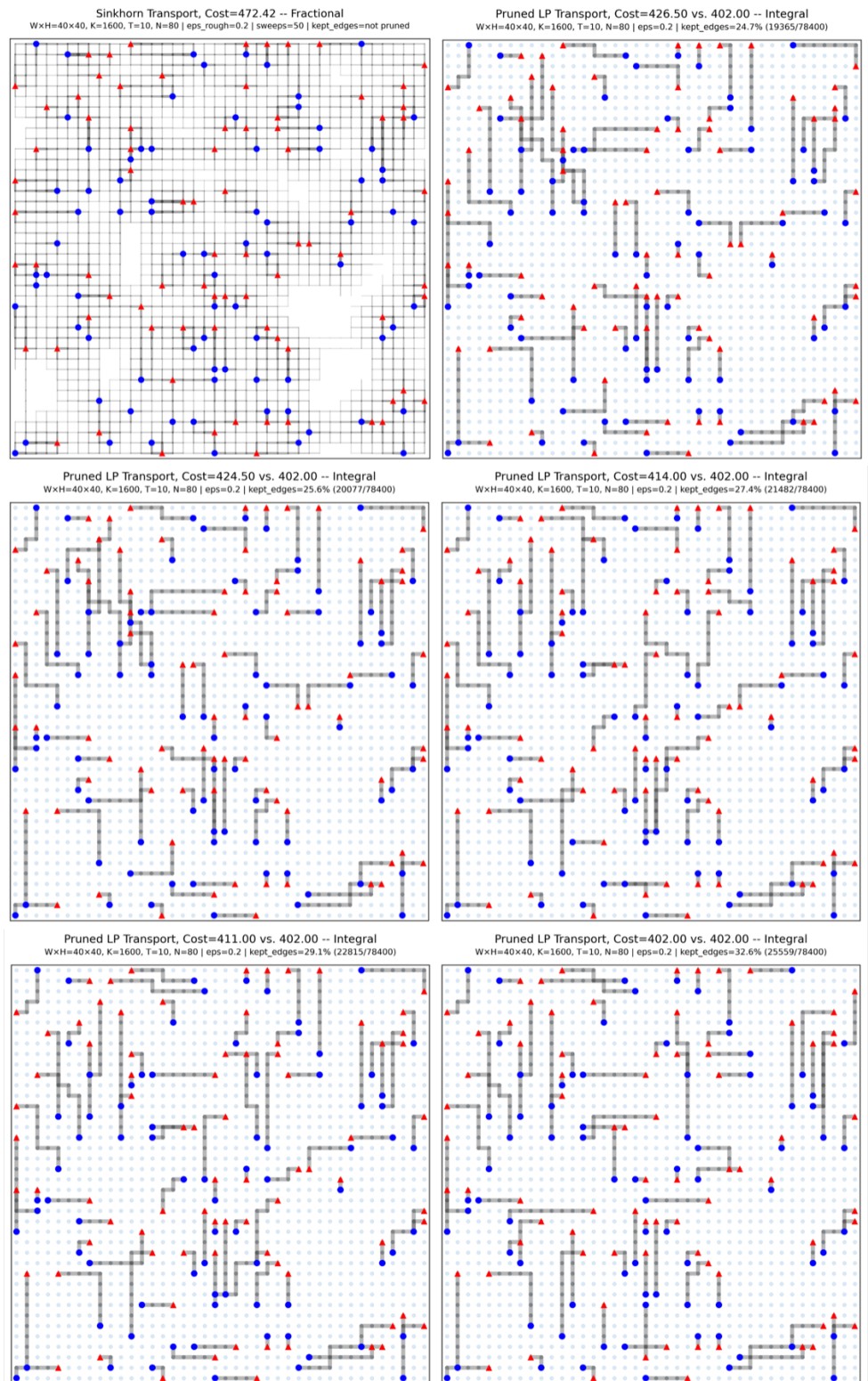

*Figure 16.* A $40 \times 40$ grid with 80 robots ▲, 80 targets •, and $T = 10$. The robot paths are superimposed over the horizon $T$. The first figure is the Schrödinger shadow that shows the likely mass transport obtained by solving **P2** after $\bar{\tau}$ Sinkhorn iterations in Appendix G; $\varepsilon = 0.2, \bar{\tau} = 50$. The next figures show the integral projection **P3** by keeping the highest-valued $X\%$ of edges from the shadow. Cost is compared with the optimal in the figure title, and the optimal min-cost transport can be achieved with $32.6\%$ edges. It is interesting to note the difference in the robot trajectories over these solutions.

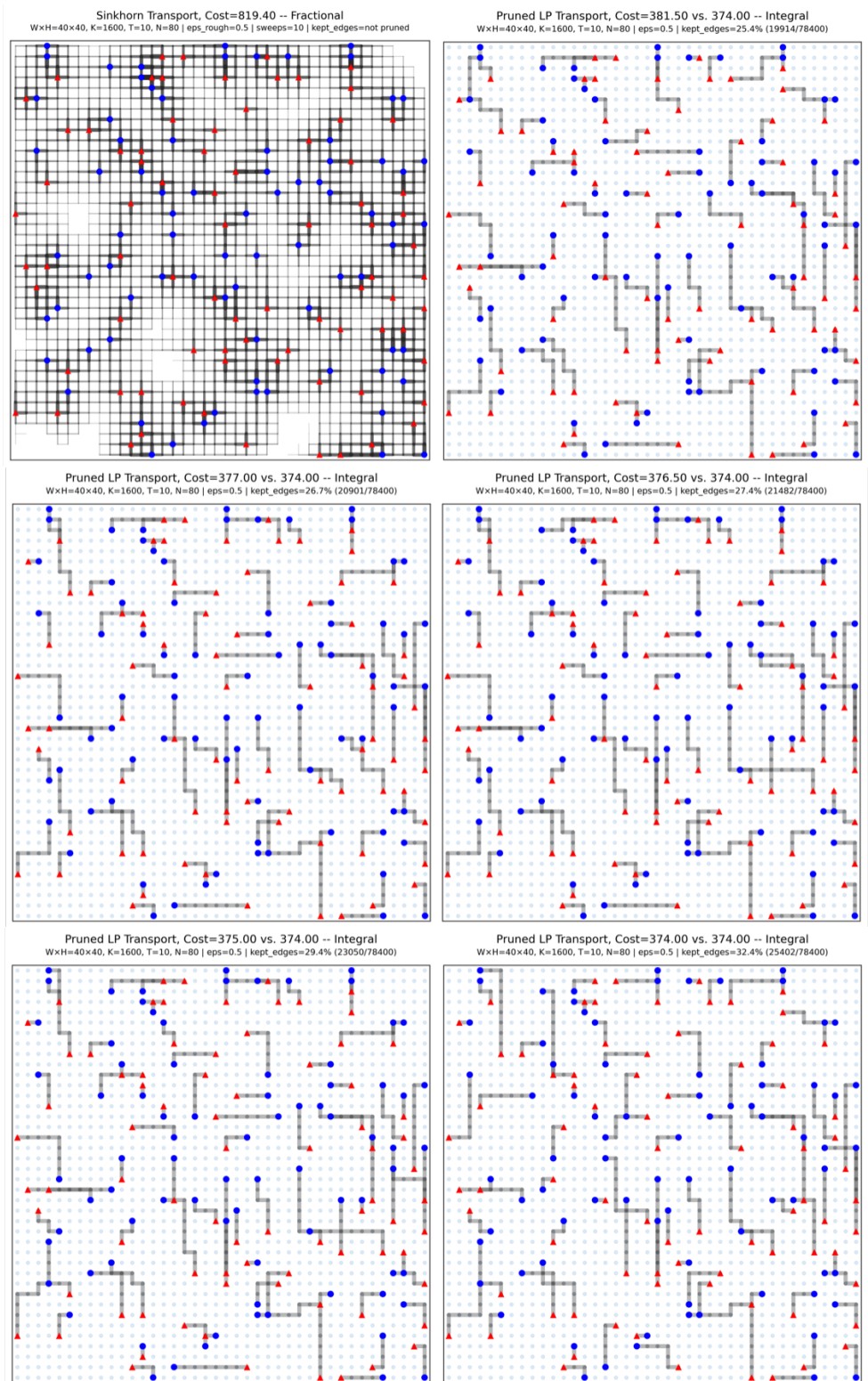

*Figure 17.* A $40 \times 40$ grid with 80 robots ▲, 80 targets ●, and $T = 10$. The robot paths are superimposed over the horizon $T$. The first figure is the Schrödinger shadow that shows the likely mass transport obtained by solving **P2** after $\bar{\tau}$ Sinkhorn iterations in Appendix G; $\varepsilon = 0.5, \bar{\tau} = 10$. The next figures show the integral projection **P3** by keeping the highest-valued $X\%$ of edges from the shadow. Cost is compared with the optimal in the figure title, which can be achieved with $32.4\%$ edges.

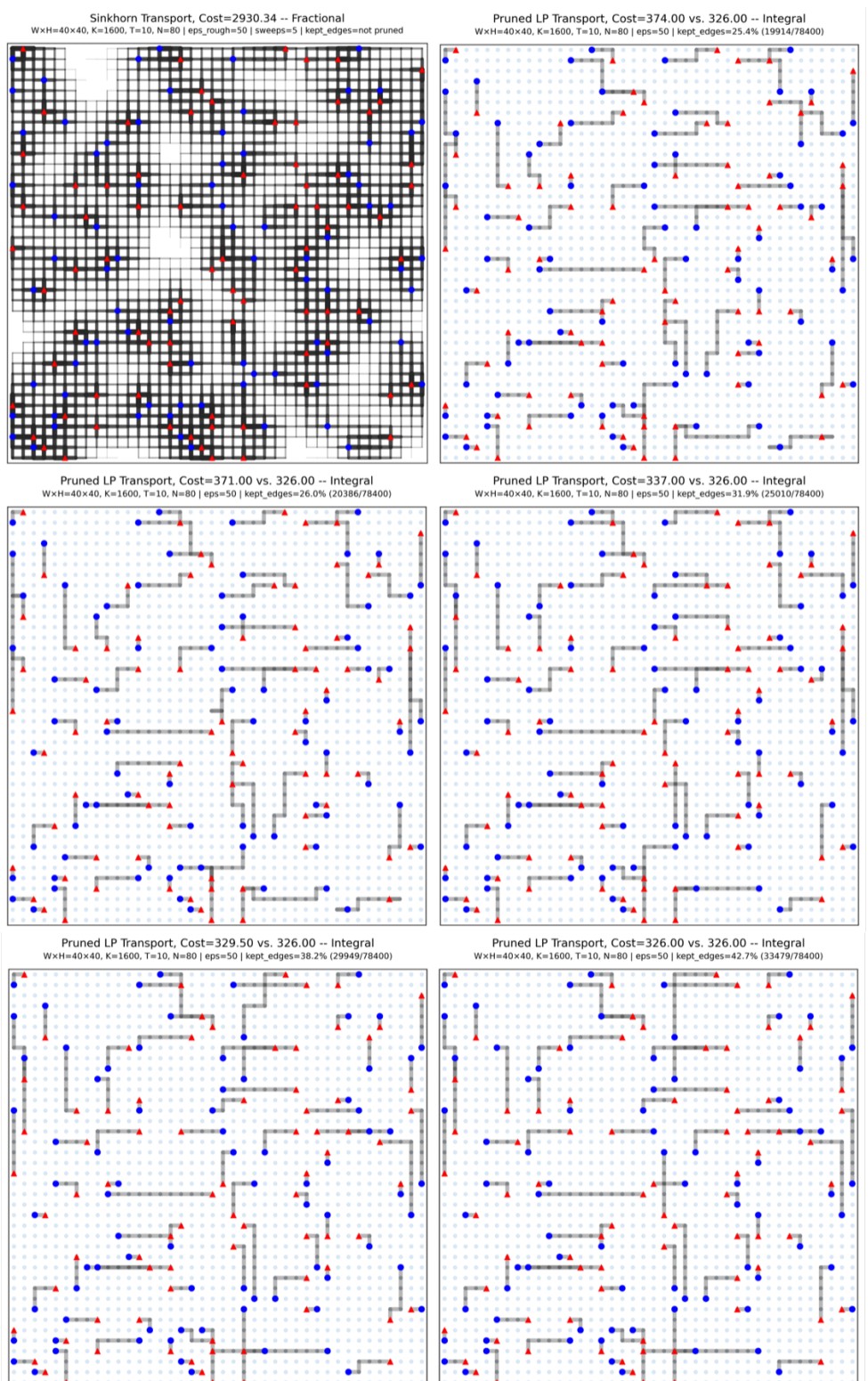

*Figure 18.* A $40 \times 40$ grid with $80$ robots ▲, $80$ targets ●, and $T = 10$. The robot paths are superimposed over the horizon $T$. The first figure is the Schrödinger shadow that shows the likely mass transport obtained by solving **P2** after $\bar{\tau}$ Sinkhorn iterations in Appendix G; $\varepsilon = 50, \bar{\tau} = 5$. The next figures show the integral projection **P3** by keeping the highest-valued $X\%$ of edges from the shadow. Cost is compared with the optimal in the figure title, which can be achieved with $42.7\%$ edges.

