# OpenReview forum: "Optimal and Scalable MAPF via Multi-Marginal Optimal Transport and Schrödinger Bridges"
_ICML.cc/2026/Conference — ICML 2026 spotlight_

### Official Review · Reviewer_enZM · 2026-03-02

**Soundness:** 2
**Presentation:** 2
**Significance:** 3
**Originality:** 3
**Overall Recommendation:** 4
**Confidence:** 3

**Summary:**

This paper studies anonymous multi-agent path finding (MAPF), where a set of robots must move from a set of start locations to a set of goal locations on a graph, but the robots are interchangeable and do not have preassigned individual targets. The paper proposes a new formulation of this problem through the lens of multi-marginal optimal transport (MMOT) on a time-expanded graph. Instead of directly searching over discrete paths for labeled agents, the method represents the evolution of the robot population through a sequence of transport plans across time, with constraints enforcing flow conservation, start/goal marginals, and collision-avoidance structure. The paper’s first main contribution is to show that the anonymous MAPF problem can be written as a linear program over these time-indexed transport matrices. Under the assumptions introduced in the paper, the authors argue that the feasible polytope has an integral structure, so solving the LP yields non-fractional robot motions and thus an optimal anonymous MAPF solution. The paper provides theoretical discussion and experimental illustrations to support the framework. The theoretical part focuses on integrality, collision-freeness, and the relation between the proposed cost design and makespan optimality, while the experiments aim to show that the reduced problem obtained from the shadow transport can preserve near-optimal solution quality while significantly shrinking the search space. Comparisons with other approach solving MAPF are not presented, although some of them are mentioned in the Related works section.

**Compliance With Llm Reviewing Policy:**

Affirmed.

**Final Justification:**

The rebuttal provides useful clarification on the intended assumptions and improves the conceptual positioning of the work relative to prior formulations. However, my main concerns remain only partially addressed: the empirical validation is still limited and lacks comparison against established MAPF baselines, the scalability/complexity discussion remains too generic, and several claims continue to depend on fairly restrictive assumptions whose practical scope is not fully substantiated. These are central issues that would require significant additions to the paper rather than a short rebuttal.

**Key Questions For Authors:**

1. Can the authors discuss the practical implications of Assumptions 3.1 and 3.5 more transparently?
2. Can the authors better position the method relative to prior anonymous/goal-invariant MAPF and provide proof (empirical/theoretical) why the proposed method should be implemented rather than the previous?

**Limitations:**

The authors have NOT discussed the limitations.
The paper has a promising and genuinely interesting core idea, and the anonymous MAPF–MMOT perspective could be worth building on. However, the current version is not yet strong enough: limited experimental validation, a generic rather than sharp complexity analysis, and issues with presentation/theoretical clarity that should be resolved before the work can be reliably built upon by others.

**Strengths And Weaknesses:**

# Soundness
## Strengths
Clear and intellectually coherent technical direction. Modeling anonymous MAPF through a transport-based view is a meaningful abstraction, and the progression from an exact LP formulation to a regularized approximation and then to a reduced LP is logically organized. The optimization pipeline is easy to follow at a high level: (P1) provides the exact formulation, (P2) produces a smooth shadow transport, and (P3) uses that shadow to sparsify the problem while aiming to preserve solution quality. The paper does not remain purely algorithmic or heuristic; it attempts to provide a theoretical foundation for the proposed framework. The chosen tools are appropriate for the intended goals. Using LP for anonymous flow-like coordination is natural, and using entropic regularization/Sinkhorn-style ideas as a scalable approximation mechanism is methodologically sensible. The support-pruning idea in (P3) is also a reasonable way to exploit structure from the relaxed solution instead of solving the full dense problem directly. Finally, the paper includes empirical evidence to show the feasibility of the approach and further proves that shadow transport can be used to reduce the search space while keeping the final cost close to the exact LP solution in the demonstrated cases. The experiments do at least match the intended role of (P2) and (P3), rather than testing something unrelated to the actual contribution.
## Weaknesses
1.	While the theoretical direction is plausible, several arguments are presented too quickly, and some assumptions are quite strong. For example, collision-freeness is established only under a carefully engineered graph model and cost structure, rather than emerging from a more general formulation. Also in my opinion, this is acceptable because this assumption matches the topology in the grid map with robot size and target size equal to 1 cell. However, the makespan-related result relies on a specific exponential time-weighted cost design, which is mathematically usable but also quite restrictive and somewhat artificial. As a result, the theoretical claims may be conditionally valid, but the paper does not always make the strength and role of these assumptions sufficiently explicit.
2.	The paper’s complexity analysis is also limited. Section 6 mainly gives a generic polynomial-time LP argument rather than a structure-aware analysis of the proposed MAPF formulation. The stated variable count scales as (K^2T), which is potentially very large, and the discussion does not clearly exploit graph sparsity to reduce the effective dimension. For (P2), the complexity discussion is even less explicit, since the paper does not provide a concrete bound in terms of the main problem parameters and instead relies on broad convergence statements and practical observations. Thus, the theoretical support for the claim of scalability remains incomplete.
3.	The experiments mainly compare (P1), (P2), and (P3) internally, which helps illustrate the paper’s own mechanism, but this is not enough to establish strong empirical support for the broader claims. There is no direct comparison with established anonymous MAPF or flow-based baselines, and the reported evaluation is limited in breadth. Key quantities such as runtime scaling, memory usage, robustness across many instances, and sensitivity to regularization/pruning parameters are not explored in enough detail. Therefore, while the experiments are directionally consistent with the method, they are not yet strong enough to fully validate the practical claims.
# Presentation
## Strength
The paper is reasonably structured, and the overall progression of ideas is understandable. The narrative moves from problem formulation to exact optimization, to regularized approximation, to sparsification, and then to theory and experiments. This gives the paper a logical backbone and the main algorithmic storyline: starting with (P1) exact LP, moving to (P2) shadow transport, and ending with (P3) reduced LP. This is easy to summarize after one full read.
##  Weaknesses
1.	The role of assumptions such as 3.1 and 3.5 needs clarification and justification. The reviewer understands that Assumption 3.1 fits the problem with the topology in the grid map with robot size and target size equal to 1 cell. The assumption of a point robot might not be realistic in many applications.
2.	There are also cross-reference mistakes, nonstandard notation, and several English/wording errors in technical passages, appearing in places where the reader is trying to verify definitions, assumptions, and optimization constraints. For example: “Assumption (1)-v” in page 4; “Appendix 3.3” in page 4; and  constraints \( \Pi_t \subseteq [\tilde{\Pi}_t]_{eta} \)
3.	In terms of positioning relative to prior work, the paper does identify relevant neighboring areas, including MAPF. The related-work discussion gives context, but the manuscript should have a comparison of the proposed approach with the state-of-the-art.
4. No code or implementation instruction provided limits the reproducibility of the work.
# Significance
## Strengths
The paper addresses an important and relevant problem. Multi-agent path finding is a well-established topic with clear relevance to robotics, logistics, warehouse automation, and coordinated planning, and the anonymous setting is also meaningful because interchangeable agents arise naturally in many applications. From that perspective, the problem itself is significant.
## Weaknesses
The assumption of a point robot might not be realistic in many applications.
Also, this is not a broadly applicable machine learning method in its current form, nor does the paper demonstrate strong practical superiority over existing anonymous MAPF baselines.
The problem matters, and the transport-based viewpoint may inspire follow-up work, but the current empirical support is not strong enough to show a major step forward in practical capability yet.
# Originality
## Strengths
The core novelty appears to be the formulation of anonymous MAPF as a multi-marginal optimal transport problem over time-indexed transport plans, together with the use of an entropically regularized “shadow” solution to guide sparsification and recovery of an integral plan. This combination is interesting and, in my view, genuinely creative.
## Weaknesses
The paper is original mainly in the way it combines ideas, rather than through an entirely unprecedented problem or primitive approach. Where originality is less strong is in the separation from adjacent literature. A clearer comparison with other methods, such as flow-based methods, or the mentioned approaches in the Related Works section, is required to claim the effectiveness.

---

> ### Author Rebuttal · Authors · 2026-03-27
>
> Thank you for a careful and constructive review. This paper makes three main contributions: (i) a novel Markovian MMOT formulation of anonymous MAPF that reduces the path-space problem to a polynomial-size LP; (ii) a structural total-unimodularity (TU) result showing that the continuous LP yields integral collision-free solutions without integer programming; and (iii) a Schrödinger-bridge/Sinkhorn shadow transport used to prune the graph and then recover an exact integral solution on the reduced graph. We would like to take this opportunity to emphasize that the connection between MMOT and MAPF is not merely a reformulation; it brings the machinery of optimal transport, Schrödinger bridges, and Sinkhorn iterations to a combinatorial robotics problem, opening new research directions across both communities.
>
> We appreciate the reviewer's thorough engagement with the technical content, and are encouraged by the positive comments. We address the main concerns below.
>
> Assumptions, Grid Layout, and Point mass robots: We do not assume a grid layout or point masses. Our setup is standard as adopted by mainstream discrete MAPF papers; see the (Stern 2019) survey, (Yu & LaValle, 2013), or the recent MAPF-GPT work.
>
> Consider a continuous space abstracted by a graph. We can discretize it finer and finer to add more vertices and edges resulting in a lot of paths from robots to targets, making (anonymous) MAPF straightforward. Assumption 3.1 rules out exactly such scenarios. It says that since robots are physical, edges and vertices cannot be added arbitrarily if they result in conflicts: 3.1(i) states that self-edges are present, i.e., a robot can wait; while 3.1(ii) states that two edges ${i_1\to i_2}$ and ${j_1\to j_2}$, with four distinct endpoints, are disallowed if robots cannot travel on them without collision; similarly, an edge ${i\to j}$ is not allowed if its traversal ends up conflicting with robots not on $i$ or $j$.
>
> Under a point-mass model, all collision constraints explicitly proven in Thm 3.4 would be vacuous. Thm 3.4 rules out head-on collisions and vertex conflicts; the remaining physically impossible pattern (cycles) is eliminated by the cost hierarchy in Assumption 3.1(v). We will also add this result formally in the revision.
>
> A grid graph where a physical robot is not bigger than a cell is a sufficient instantiation, chosen for experimental clarity. However, we could replace neighbors with any adjacency structure on any finite connected graph (under 3.1(i) and (ii)) and the LP and the corresponding TU argument hold.
>
> Assumption 3.5 is used only for minimum-makespan and is not a universal condition; Lemma 3.7 then provides the practical alternative.
>
> We will make all of these points explicit in the final version.
>
> Relation to prior work: We agree that anonymous MAPF is related to classical network-flow formulations (e.g., Yu & LaValle, 2013). However, prior flow-based formulations either produce fractional solutions or require integer programming to enforce integrality. Our contribution is to develop a Markovian MMOT viewpoint and show that under mild conditions the resulting continuous LP is integral via TU, for which no rounding or branch-and-bound is needed. This MMOT foundation further enables the Schrödinger-bridge interpretation and the shadow-transport pruning pipeline, which have no counterpart in prior flow-based MAPF work.
>
> Empirical validation: We do agree that the reviewer raised important points regarding comparative studies, scalability, and sensitivity to pruning parameters. We are committed to extending the current experimental setup significantly; our detailed plan, including initial scaling results, and parameter insights are described in our response to Reviewer 1mfi under Comparison with MAPF solvers and Scalability, grid environments, and pruning sensitivity.
>
> Complexity: On sparse graphs, the effective problem size scales with feasible transitions, roughly $\mathcal O(|\mathcal E|T)$, rather than dense $\mathcal O(|\mathcal V|^2 T)$. We will provide explicit complexity statements in terms of graph sparsity and specific LP solvers in the revision. For Sinkhorn, we do not provide any explicit convergence statement but rely on the literature where multi-marginal Sinkhorn is shown to be linear. See also our plan for more experiments, as mentioned above, which will strengthen Section 6 further.
>
> Finally, we appreciate the comments on presentation and limitations. A key limitation is that the TU guarantee applies to the anonymous setting and extending it to labeled MAPF remains open. We take the reviewer's feedback seriously and are committed to strengthening the paper on all fronts noted above. We will tighten notation and cross-references, correct language issues, and make scope and limitations more explicit. We will be happy to provide code files with the revision.

---

> > ### Author Rebuttal · Reviewer_enZM · 2026-04-02
> >
> > The rebuttal is thoughtful and mostly legitimate in tone, but it is much stronger as a roadmap for revision than as an actual resolution of the current paper’s weaknesses. The rebuttal provides useful clarification of the intended assumptions and improves the work's conceptual positioning relative to prior formulations. However, my main concerns remain only partially addressed: the empirical validation is still limited and lacks comparison against established MAPF baselines, the scalability/complexity discussion remains too generic, and several claims continue to depend on fairly restrictive assumptions whose practical scope is not fully substantiated. These are central issues that would require significant additions to the paper rather than a short rebuttal.

---

> > > ### Author Response · Authors · 2026-04-04
> > >
> > > Since the last response, we have significantly expanded the empirical validation with over 460 new experiments (Gurobi, 2022 MacBook) and address the three remaining concerns with concrete evidence below.
> > >
> > > 1. Empirical validation and scaling. We present 162 new experiments, ranging from 369K to 3.4M vars, where we ran P1 and the P2+P3 pipeline across 8 grid sizes at 5% robot density ($N = 0.05K$), each averaged over 14-25 independent random instances ($T=30$, $\varepsilon=0.2$):
> > >
> > > $K=2{,}500$ ($N=125$, 20 runs): P1 = 15s (369K vars), P2+P3 = 4s (120K vars), 3.6x speedup, 8.0% gap
> > >
> > > $K=5{,}625$ ($N=281$, 25 runs): P1 = 55s (835K), P2+P3 = 11s (288K), 5.0x, 8.7% gap
> > >
> > > $K=8{,}100$ ($N=405$, 19 runs): P1 = 103s (1.2M), P2+P3 = 18s (441K), 5.7x, 9.0% gap
> > >
> > > $K=10{,}000$ ($N=500$, 24 runs): P1 = 132s (1.5M), P2+P3 = 26s (605K), 5.1x, 5.9% gap
> > >
> > > $K=13{,}225$ ($N=661$, 14 runs): P1 = 193s (2M), P2+P3 = 33s (775K), 5.8x, 8.1% gap
> > >
> > > $K=15{,}625$ ($N=781$, 23 runs): P1 = 257s (2.3M), P2+P3 = 48s (991K), 5.3x, 5.9% gap
> > >
> > > $K=19{,}600$ ($N=980$, 18 runs): P1 = 364s (2.9M), P2+P3 = 62s (1.2M), 5.8x, 7.2% gap
> > >
> > > $K=22{,}500$ ($N=1{,}125$, 19 runs): P1 = 478s (3.4M), P2+P3 = 67s (1.4M), 7.1x, 6.5% gap
> > >
> > > P1 runtime scales as $K^{1.68}$ and P2+P3 as $K^{1.15}$ (nonlinear least squares, $aK^p + b$, on 162 runs; $R^2 = 0.96$ and $0.74$). The speedup grows with problem size (3.6x at $K=2{,}500$ to 7.1x at $K=22{,}500$). The cost gap is consistently below 10% (median 6.4%) while retaining only 32-43% of edges. Since P1 already provides the exact optimum, P2+P3 offers a practical speed-quality tradeoff: a 5-7x speedup at under 10% cost degradation, with the user free to choose P1 when optimality is required.
> > >
> > > Every single solution across all 460+ runs (scaling, sensitivity, non-uniform, and baseline) was verified integral. For reference, the largest instance in the original submission was $K = 1{,}600$; the scaling study now starts at $K=2{,}500$ and reaches $K = 22{,}500$, a 14x increase in problem size.
> > >
> > > 2. Sensitivity to P2+P3 parameters: In our response to Reviewer UTFA, we provide a detailed experiment that sweeps $(\varepsilon, \lambda)$ over 13 instances at $K = 10{,}000$ (1.5M variables). We verify that Sinkhorn convergence speed scales inversely with $\varepsilon$ as expected from the entropic regularization literature, with the cost-quality tradeoff controlled primarily by $\varepsilon$. We also validate the pipeline under non-uniform terrain costs (24 instances with 1.5M variables each, 5.1% gap, 5.4x speedup), confirming that performance does not degrade.
> > >
> > > 3. Complexity on sparse graphs. The reviewer correctly noted that $K^2T$ is the dense variable count. On grid graphs, each vertex has $\leq 5$ neighbors (4 cardinal + self-loop), so the actual variable count is $|E|T \approx 5KT$, not $K^2T$. For example, at $K=22{,}500$ with $T=30$: $|E|T = 3{,}357{,}000$ vs. $K^2T = 1.5 \times 10^{10}$ as reflect in the scaling studies. On general graphs, the effective size scales as $\mathcal{O}(|\mathcal{E}|T)$ where $|\mathcal{E}|$ is the edge set of $\mathcal{G}$, not $|\mathcal{V}|^2$.
> > >
> > > 4. Comparison with prior anonymous MAPF. A direct comparison with a closely related anonymous MAPF baseline (Ma & Koenig, 2016) is reported in our response to Reviewer 1mfi.
> > >
> > > We note: (a) P1 LP solves 22,500-vertex instances in under 8 mins on a laptop; (b) all 460+ solutions are integral without integer constraints. (c) The resulting P2+P3 pipeline further reduces solve time to ~67 seconds ($K=22{,}500$) with <7% cost degradation, scaling nearly linearly in $K$.
> > >
> > > We are not aware of any anonymous MAPF method that provides polynomial-time integrality guarantees from first principles and scales almost linearly in practice via a principled Schrödinger/MMOT and Sinkhorn pipeline.
> > >
> > > 5. Assumptions and limitations. Our assumptions are standard in the MAPF literature (Stern 2019; Yu & LaValle 2013; Standley 2010) and apply to any finite connected graph, not only grids; the grid+obstacle setting is chosen for experimental clarity. Table 1 from (Ma & Koenig, 2016), quoted in our response to Reviewer 1mfi, uses the exact same grid + obstacle structure for a comparison with related methods.
> > >
> > > An important limitation is that TU applies to the anonymous setting; extending to labeled MAPF (fixed robot-target pairings) breaks TU and requires an ILP.
> > >
> > > All 460+ experiments reported above are complete. We will include the full scaling table, graphs, power-law fits, sparsity-aware complexity, sensitivity analysis, and non-uniform terrain experiments. We will also provide the code for full reproducibility.
> > >
> > > We thank the reviewers for their constructive engagement. The questions raised helped us improve the content, and the experimental pipeline we have built in response (over 460 new experiments across scaling, sensitivity, non-uniform costs, and baseline comparisons, all on a single 2022 MacBook) has significantly improved this work.

---

### Official Review · Reviewer_UTFA · 2026-03-13

**Soundness:** 3
**Presentation:** 3
**Significance:** 3
**Originality:** 3
**Overall Recommendation:** 4
**Confidence:** 4

**Summary:**

The paper studies anonymous multi-agent path finding (MAPF), where robots shoud reach different targets on a graph without collisions while minimizing cost. The problem is  formulated as a multi-marginal optimal transport (MMOT) problem with the idea that its LP relaxation will be  totally unimodular. This unimodularity guarantees the integral collision-free paths without integer programming.
To improve scalability, the authors use Schrödinger bridge that is based on the idea of  entropic regularization and provide the Sinkhorn algorithm iterations to compute a probabilistic shadow transport plan to prune the unlikely edges. Experiments show that this pruning significantly reduces computation while retaining near-optimal solutions.

**Compliance With Llm Reviewing Policy:**

Affirmed.

**Key Questions For Authors:**

1. The paper relies on Assumption 3.1 and other related structural  conditions for total unimodularity and integral solutions.
Could the authors clarify the practicality and the validity of these assumptions in real-world MAPF scenarios with obstacles or complex graphs?
If these assumptions are often violated, what is the expected impact on solution integrality and optimality?

2. The pruning approach based on Schrodinger bridge involves parameter choices epsilon, lambda, eta, and delta. Can you provide the explaionation on how to select these parameters in practice, and whether the approach is sensitive to these choices?

3. The experimental results focus largely on grid graphs with uniform costs and simple obstacle settings. Will the performacnce degrade in case of non-uniform costs?

**Limitations:**

The limitations are mentioned in the paper, however, there are some limiations such as mentioned in the questions section. Addressing them would be beneficial for the brader domain.

**Strengths And Weaknesses:**

Soundness:  The paper is technically sound, and it is supported by both theoretical and empirical evaluation. The authors first prove some of the key properties such as total unimodularity. This ensures the integral solutions to the MAPF formulation without explicit integer constraints.
The use of Schrödinger bridge regularization with Sinkhorn algorithm iterations is a well known methods in the optimal transport methods literature.

Presentation: The paper is clearly written and well structured. If I rembere correctly, I have reviewed this paper in the past, and there are significant differences in the presentations and writing. The paper is well positioned and I am happy to see the changes.

Significance: The scalable, collision-free path planning for anonymous agents isn an important problem in multi-agent systems and robotic. Connecting the MAPF with MMOT, the paper provides both theoretical insight and practical algorithms.

Originality:
The paper introduces a novel formulation of anonymous MAPF as a Markovian MMOT problem with a polynomial-size LP. Further, this LP has an integral solutions. Though, it is commonly used in the literature,  the combination of Schrödinger bridge based entropic regularization with a shadow transport is a nice approach to solve the problem.

---

> ### Author Rebuttal · Authors · 2026-03-27
>
> Thank you for the positive review. We are glad that the MMOT–MAPF connection, TU-based integrality, and the Schrödinger bridge pruning pipeline came across clearly. We address the reviewer's questions below.
>
> Practicality of Assumption 3.1 and Non-Uniform Costs: Assumption 3.1 is standard in the discrete MAPF literature (Stern, 2019; Yu \& LaValle, 2013). Consider a continuous space abstracted by a graph. We can discretize it finer and finer to add more vertices and edges resulting in a lot of paths from robots to targets, making (anonymous) MAPF straightforward. Assumption 3.1 rules out exactly such scenarios. It says that since robots are physical, edges and vertices cannot be added arbitrarily if they result in conflicts: 3.1(i) states that self-edges are present, i.e., a robot can wait; while 3.1(ii) states that two edges ${i_1\to i_2}$ and ${j_1\to j_2}$, with four distinct endpoints, are disallowed if robots cannot travel on them without collision; similarly, an edge ${i\to j}$ is not allowed if its traversal ends up conflicting with robots not on $i$ or $j$.
>
> Assumptions 3.1(iii-v) are logical edge-cost assumptions that are structural and always required. In particular, 3.1(iii) states that for every edge in the graph the cost to travel it is finite; this avoids the scenario where the graph is connected but infinite cost effectively disconnects it. 3.1(iv) states that the cost to travel a path equals the sum of the costs to travel the edges along that path. Finally, 3.1(v) states that waiting at a vertex is strictly cheaper than moving, i.e., moving expends more energy than staying put.
>
> Violations of the graph-edge assumptions (i) and (ii) will result in conflicts that are invisible to the LP; in some sense, this is not a limitation of the formulation but rather an incomplete graph abstraction that does not capture the physical conflicts, and the modeling must be improved. The cost assumptions are basic and their violation triggers erratic robot behavior, e.g., a robot may move to a neighboring location and then return because moving was cheaper than waiting at its own location.
>
> The proposed formulation fully supports non-uniform, time-varying, nonnegative costs. Assumption 3.1(v) only requires that waits are cheaper than moves and moves have finite cost. The actual values can vary arbitrarily across edges and time. The LP and TU guarantees do not depend on the cost magnitudes, but only on the constraint matrix. There will therefore be no degradation with non-uniform costs. Note however that choosing costs arbitrarily may result in undesirable robot behavior; some conditions for well-behaved trajectories (no oscillations, temporal urgency, temporal subadditivity) are described after Theorem 3.4.
>
> Parameter Sensitivity: This is a great suggestion and we will expand on this in the revision. As we noted briefly in the remarks in Section 5, the parameters straddle the spectrum from optimal to scalable. Setting ${\lambda = \eta = 0}$ recovers the exact P1 from P3 regardless of $\varepsilon$. In practice: $\varepsilon$ controls the smoothness of the shadow transport (smaller $\varepsilon$ leads to sharper P2 transport that is closer to P1; larger $\varepsilon$ leads to smoother P2 transport, faster Sinkhorn convergence); $\eta$ controls the pruning threshold (larger $\eta$ results in more aggressive pruning, and therefore smaller LP); $\lambda$ controls the bias toward the shadow in the modified P3 cost; and $\delta$ is a small numerical safeguard.
>
> Sensitivity to $(\varepsilon, \bar{\tau})$ was explored in Appendix H (Figs. 11-13) across ${(0.2, 50), (0.5, 10), (50, 5)}$. In general, the approach is sensitive to a large $\varepsilon$ as it puts equal mass on all edges making pruning ineffective; in other words, $\varepsilon$ controls the spread and uniformity of the shadow. On the other hand, $\lambda$ controls how closely the resulting P3 transport follows the shadow: a large $\lambda$ penalizes edges with small shadow flow, effectively forcing P3 to follow the shadow transport. If the costs are uniform, $\varepsilon$ and  $\lambda$ provide a beneficial tiebreaker among edges that are otherwise equivalent in cost, favoring those with a darker shadow. The parameter $\delta$ is a small constant that regularizes the logarithm and has negligible effect on the solution. We will expand this study by varying $\lambda$, $\eta$, and $\delta$ independently and reporting on how to select these parameters in practice, feasibility thresholds, and cost degradation across multiple random instances.
>
> We also refer the reviewer to our response to Reviewer 1mfi, where we report scaling results up to 22,500 vertices (3.4M variables) on a 2022 MacBook; we will add comprehensive large-scale scaling and comparison experiments in the revision. Finally, we thank the reviewer for the positive assessment and the constructive questions. We will be happy to provide code files with the revision.

---

> > ### Author Rebuttal · Reviewer_UTFA · 2026-04-03
> >
> > Some of my points are addressed. However, given the changes required particularly regarding the usefulness and need of various assumptions, the current work require more discussion. Further, the sensitivity of various parameters requires more thought. As per the rebuttal I see only some immediate thoughts, but in general it requires more brainstorming. As of now I am retaining the score. Looking forward to more insightful comments on this.

---

> > > ### Author Response · Authors · 2026-04-04
> > >
> > > Since the last response, we have conducted a systematic scaling study (162 new experiments across 8 grid sizes reported in our response to Reviewer enZM). A direct comparison with a related anonymous MAPF baseline (Ma & Koenig, 2016) is reported in our response to Reviewer 1mfi. The reviewer's remaining concerns on parameter sensitivity and non-uniform costs are discussed below.
> > >
> > > 1. Parameter sensitivity: We swept $(\varepsilon, \lambda)$ over 13 independent random instances on $K = 10{,}000$ vertices ($N = 500$ robots, $T = 30$, 1.5M variables in the P1 LP), with 5 values of $\varepsilon$ and 4 values of $\lambda$ for a total of 260 runs.
> > >
> > > P1 baseline: avg cost = 2904, avg time = 130s.
> > >
> > > Cost gap (%) relative to P1:
> > >
> > > $\varepsilon \backslash \lambda$ | 0 | 0.5 | 1.0 | 5.0
> > >
> > > 0.1 | 2.3 | 2.5 | 2.7 | 3.0
> > >
> > > 0.2 | 4.3 | 5.0 | 5.8 | 7.3
> > >
> > > 0.5 | 11.1 | 12.7 | 14.0 | 16.5
> > >
> > > 1.0 | 17.3 | 18.9 | 20.1 | 23.1
> > >
> > > 5.0 | 17.1 | 18.0 | 18.7 | 19.9
> > >
> > > Sinkhorn time (seconds) per $\varepsilon$:
> > >
> > > 0.1: 20s (307 sweeps);
> > >
> > > 0.2: 9s (132 sweeps);
> > >
> > > 0.5: 7s (97 sweeps);
> > >
> > > 1.0: 5s (75 sweeps);
> > >
> > > 5.0: 3s (39 sweeps).
> > >
> > > P2+P3 time with $\lambda = 0$ are:
> > >
> > > 0.1: 48s;
> > >
> > > 0.2: 26s;
> > >
> > > 0.5: 22s;
> > >
> > > 1.0: 23s;
> > >
> > > 5.0: 25s; [higher runtime since a diffusive shadow makes pruning ineffective, ending in more variables kept.]
> > >
> > > Parameter significance: The results reveal a clear structure.
> > >
> > > $\varepsilon$ controls the shadow sharpness and is the dominant parameter. Small $\varepsilon$ ($\leq 0.2$) produces a concentrated shadow close to the P1 optimum, enabling aggressive pruning with low cost gap (3-5%). However, smaller $\varepsilon$ requires more Sinkhorn iterations to converge: 307 sweeps for $\varepsilon = 0.1$ vs. 132 for $\varepsilon = 0.2$ vs. 39 for $\varepsilon = 5.0$. The best time-quality tradeoff is at $\varepsilon = 0.2$, which achieves 4-7% gap in ~26 seconds (5.0x speedup over P1 at 130s).
> > >
> > > Sinkhorn convergence speed scales inversely with $\varepsilon$, as expected from the entropic regularization methodology. However, the cost gap increases as $\varepsilon$ increases (from 2-3% at $\varepsilon = 0.1$ to 17-20% at $\varepsilon = 5.0$), reflecting the smoothing effect that makes the shadow less discriminative and the pruned graph less targeted.
> > >
> > > $\lambda$ controls the bias toward the shadow in the P3 cost. Its effect is mild: increasing $\lambda$ from 0 to 5 adds roughly 1-6% to the gap depending on $\varepsilon$. More broadly, $\lambda$ and $\varepsilon$ together add robustness to the pipeline: the P3 cost is not determined by the original edge costs alone but is adjusted in line with the shadow, consistent with the role of entropic regularization in smoothing combinatorial objectives.
> > >
> > > $\delta$ is a numerical safeguard for the logarithm in the modified P3 cost.
> > >
> > > We will add the corresponding sensitivity plots across graph sizes, scaling trends, and power-law fits in the final version.
> > >
> > > 2. Practical parameter choice: In all the experiments we have run for the rebuttal, over 460, with variables ranging from 369K to 3.4M (2,500 to 22,500 vertices), the following is a safe choice and almost always works without much fine-tuning: $\delta = 10^{-6}$, $\varepsilon = 0.2$, stop Sinkhorn when the last 10 iterates vary by < 1%, while $\lambda = 0$ produces the best cost gap. For the pruning threshold $\eta$, we have found that retaining 40-48% of edges (depending on $\varepsilon$) consistently yields feasible P3 solutions; we kept it around 40-45% for the above $K = 10{,}000$ sensitivity study.
> > >
> > > 3. Non-uniform costs. In this experiment, we assign each cell a random arrival cost $\sim \text{Uniform}[0.6, 1]$, choose wait costs to be less than moves $\sim \text{Uniform}[0.1, 0.5]$, with wait at target $= 0$. This scenario models e.g., uneven terrains; note that this simple strategy preserves Assumption 3.1(v). Following the parameter recipe from 2. above, we ran 24 independent instances at $K = 10{,}000$:
> > >
> > > P1 avg time = 138s, P2+P3 avg time = 25s, avg gap = 5.1%, speedup = 5.4x.
> > >
> > > Every solution was verified integral. For reference, the uniform-cost baseline gives gap = 4.3% and speedup = 5.0x. The gap and speedup under non-uniform costs are comparable with ($\varepsilon=0.2, \lambda=0$) results above and scaling experiments from enZM response, confirming that the pipeline adapts to the cost landscape.
> > >
> > > 4. Comparison with prior work. Please see our response to Reviewer 1mfi.
> > >
> > > All 460+ experiments reported above are complete. We will include the full scaling table, graphs, power-law fits, sparsity-aware complexity, sensitivity analysis, and non-uniform terrain experiments. We will also provide the code for full reproducibility.
> > >
> > > We thank the reviewers for their constructive engagement. The questions raised helped us improve the content, and the experimental pipeline we have built in response (over 460 new experiments across scaling, sensitivity, non-uniform costs, and baseline comparisons, all on a single 2022 MacBook) has significantly improved this work.

---

### Official Review · Reviewer_1mfi · 2026-03-16

**Soundness:** 3
**Presentation:** 3
**Significance:** 3
**Originality:** 3
**Overall Recommendation:** 5
**Confidence:** 2

**Summary:**

Multi-agent path finding (MAPF) studies how a group of robots can move from their starting locations to target locations on a shared graph without colliding while minimizing travel cost or time. The problem is combinatorial because assignment, routing, and scheduling decisions are tightly coupled. Prior work has explored search-based methods and network-flow formulations, but these approaches do not explicitly characterize when linear programming relaxations yield integral solutions or connect MAPF to modern optimal transport frameworks.

The paper proposes a new formulation of anonymous MAPF through multi-marginal optimal transport (MMOT) defined over path space. By exploiting the Markov structure of robot motion, the authors reduce the exponentially large MMOT tensor into a polynomial-size linear program over pairwise transports between consecutive time steps. They show that, under certain structural assumptions, the resulting LP has a totally unimodular constraint matrix, which guarantees integral and collision-free solutions without integer programming. To address scalability, the paper further introduces a probabilistic relaxation using Schrödinger bridges that leads to an entropic regularization solved with Sinkhorn-style iterations. The resulting fractional transport (shadow transport) is then used to prune the graph and guide a reduced LP that recovers integral robot trajectories.

The experiments evaluate the proposed framework on grid-based environments with obstacles. The authors compare the exact LP formulation, the fractional Sinkhorn relaxation, and the final projected solution on the pruned graph. Results show that the shadow transport often concentrates mass on a small subset of edges, allowing the algorithm to remove roughly 60–80% of edges while maintaining near-optimal costs. Visual examples demonstrate that the final projected solution closely matches the optimal LP solution, with only minor cost degradation in larger instances.

**Compliance With Llm Reviewing Policy:**

Affirmed.

**Final Justification:**

The new experiments add much value to the paper and resolves my main concerns.

**Key Questions For Authors:**

* How does the proposed approach compare in runtime and solution quality against established MAPF solvers?
* How sensitive is the pruning strategy to the parameters used in the entropic regularization and edge-thresholding steps?

**Limitations:**

Yes

**Strengths And Weaknesses:**

### Strengths
* The paper provides a clean and conceptually appealing connection between multi-agent path finding and multi-marginal optimal transport.
* The LP formulation with total unimodularity guarantees integral and collision-free trajectories without requiring integer programming, which is a strong structural result.

### Weaknesses
* The experimental evaluation is relatively limited and does not compare the method with standard MAPF solvers such as CBS or other state-of-the-art approaches.
* The experiments are restricted to grid environments and relatively modest problem sizes, which makes it difficult to assess practical scalability.

---

> ### Author Rebuttal · Authors · 2026-03-27
>
> Thank you for the positive and thoughtful review. We are glad that the MMOT formulation and the TU-based integrality result came across as intended. This paper is primarily theoretical: all claims are complete and explicitly proven, except for the P3 vs. P1 gap, which we demonstrated empirically. We address the reviewer's concerns below.
>
> Comparison with MAPF solvers: Search-based solvers like CBS require an explicit target assignment and do not exploit the structure of anonymous MAPF, where optimal assignment emerges for free from the transport plan with polynomial-time and integrality guarantees. In contrast, our LP solves assignment, routing, and collision avoidance in a single shot, with TU guaranteeing integrality and polynomial complexity from first principles. This is the reason we focused on validating the theoretical claims (integrality, collision-freeness, shadow-transport pruning) rather than runtime benchmarks compared with heuristics. That said, we agree with the reviewer that comparative studies would improve the paper and would be happy to add relevant experiments in the revision.
>
> Scalability, grid environments, and pruning sensitivity: We do not assume grid graphs. From our initial submission, the experiments in Figs. 1, 4, and 5 do not assume regular grid structure because of quite a large number of randomly placed obstacles, which increase the effective graph diameter well above the regular grid setting, the node degrees are not uniform, and the obstacles create narrow corridors and choke points. The LP formulation and TU guarantee are completely general and only require Assumption 3.1, not any geometric or structural property of grids. We have detailed the consequences of this assumption in our response to Reviewer enZM.
>
> The setup adopted in this paper is standard and widely adopted in mainstream MAPF papers. Grids with obstacles are one sufficient instantiation, chosen for experimental clarity. On sparse graphs, the effective problem size scales as $\mathcal{O}(|\mathcal{E}|T)$ rather than $\mathcal{O}(|\mathcal{V}|^2 T)$.
>
> As the reviewer noted, our experiments were conducted on medium-sized problems. The LP formulation scales polynomially and is limited in practice only by machine resources. During preparation of this rebuttal, we switched to Gurobi on the same 2022 MacBook. From our initial submission (scipy linprog), a problem with 1,600 vertices ($40\times 40$ grid with $T=10$ and 80 robots) took about $2$ seconds; with Gurobi, the same run on average is $0.5s$.
>
> For much larger graphs and $T=30$, we have these from Gurobi: 2,500 vertices (369K variables, $50\times 50$, 100 robots) in $14s$; 10,000 vertices ($1.5M$ vars over a $100\times 100$, 200 robots) in $102s$; 15,625 vertices ($2.3M$ vars over a $125\times 125$, 250 robots) in $142s$; and 22,500 vertices ($3.4M$ vars over a $150\times 150$, 300 robots) in $214s$. All runtimes are rounded averages over 5 runs. These scale significantly more favorably than worst-case LP complexity.
>
> We also ran scaling experiments on the P2+P3 pipeline at 5% robot occupancy on large instances (averaged over 5 independent runs, $T=30$, with Sinkhorn parameters $\varepsilon=0.2, \bar\tau=150$): $100\times 100$ (500 robots), 5.9x runtime speedup at 10% cost gap; $125\times 125$ (781 robots), 5.4x speedup at 7% gap; $150\times 150$ (1,125 robots), 6.5x speedup at 5% gap. The cost gap consistently decreases on larger instances. Further improvements via early stopping, feasibility seeding, and parameter tuning are potential extensions.
>
> We are happy to add a comprehensive scaling study along these lines and will accordingly revise Section 6.
>
> Sensitivity to $(\varepsilon, \bar{\tau})$ was explored in Appendix H (Figs. 11-13) across ${(0.2, 50), (0.5, 10), (50, 5)}$. In general, the approach is sensitive to a large $\varepsilon$ as it puts equal mass on all edges making pruning ineffective; in other words, $\varepsilon$ controls the spread and uniformity of the shadow. On the other hand, $\lambda$ controls how closely the resulting P3 transport follows the shadow: a large $\lambda$ penalizes edges with small shadow flow, effectively forcing P3 to follow the shadow transport. If the costs are uniform, $\varepsilon$ and  $\lambda$ provide a beneficial tiebreaker among edges that are otherwise equivalent in cost, favoring those with a darker shadow. The parameter $\delta$ is a small constant that regularizes the logarithm and has negligible effect on the solution. We will expand this study by varying $\lambda$, $\eta$, and $\delta$ independently and reporting on how to select these parameters in practice, feasibility thresholds, and cost degradation across multiple random instances.
>
> Finally, we appreciate the reviewer's recognition of the paper's contributions. We will be happy to provide code files with the revision.

---

> > ### Author Rebuttal · Reviewer_1mfi · 2026-04-04
> >
> > The rebuttal provides helpful clarifications, particularly regarding scalability and parameter sensitivity, and I appreciate the additional details. However, it does not fully resolve my main concerns, especially the lack of comparison with established MAPF baselines and the limited empirical validation relative to the paper’s algorithmic claims. As also noted by the other reviewers, the work presents an interesting and valuable research direction, particularly from a theoretical perspective, but would benefit from further development and more comprehensive evaluation before publication. Therefore, I will keep my original score.

---

> > > ### Author Response · Authors · 2026-04-04
> > >
> > > The reviewer's remaining concerns are the lack of comparison with MAPF baselines and limited empirical validation. Since the last response, we have conducted: 162 scaling experiments (reported in our response to Reviewer enZM); a systematic parameter sensitivity study (260 experiments) and non-uniform cost experiments (24 experiments), both reported in our response to Reviewer UTFA; and a direct comparison with a closely related anonymous MAPF baseline, which we report below.
> > >
> > > 1. Comparison with prior work. A closely related anonymous MAPF baseline is CBM (Ma & Koenig, 2016), which provides a comprehensive comparison, over multiple MAPF baselines, on 30x30 grids with 10% obstacles; the exact same grid+obstacle+cost setting adopted in our paper. Labeled MAPF solvers (CBS, LaCAM) are C++ based and solve a different problem requiring separate target assignment. However, for all practical purposes, the results quoted in Ma & Koenig, 2016, Table 1 provide a comparable benchmark (they report on 30x30 grids, 10% obstacles):
> > >
> > > CBM (method introduced by Ma & Koenig): 50 agents, 5.32s;
> > >
> > > ILP (Integer Linear Program): 40 agents, 153s (14% success within 5 mins);
> > >
> > > ILP: 50 agents, 162s (only 4% success within 5 mins).
> > >
> > > We ran 15 instances on the same setting with the same cost structure (30x30, 10% obstacles, 300 agents):
> > > P1 avg: 0.54s (optimal, integral)
> > > P2+P3 avg: 0.49s (0.63% gap)
> > >
> > > Even discounting for hardware and implementation differences, our P1 (at 10x faster on 6x more agents) is a significant speedup over CBM. Our framework already handles $K = 22{,}500$ vertices with $N = 1{,}125$ agents (3.4M variables), where P2+P3 solves in 67s, a regime no prior method has reached, and is limited only by machine resources.
> > >
> > > The ILP numbers in Table 1 tell an even more compelling story. On a $30\times 30$, 4-neighbor grid with 10% obstacles and $T=30$, the number of variables is at most $|\mathcal{E}|T < 135{,}000$. For a problem of this modest size with 50 agents, the integer program was successful only 4% of the time, and that too in 162s. Our integrality guarantee from TU means that we do not need a combinatorial IP but the LP will find integer solutions; which for the same grid with 300 agents took 0.54s. The ILP comparison illustrates the practical value of TU-based integrality.
> > >
> > > 2. Empirical validation. We kindly refer the reviewer to our responses to Reviewers enZM and UTFA, where we report: (a) 162 scaling experiments across 8 grid sizes from $K = 2{,}500$ (369K variables) up to $K = 22{,}500$ (3.4M variables) with speedups growing from 3.6x to 7.1x, where average P1 time scales as $K^{1.68}$, while average P2+P3 time scales almost linearly as $K^{1.15}$; (b) a full $(\varepsilon, \lambda)$ sensitivity study over 260 runs (13 instances, each with 20 parameter combinations) at $K = 10{,}000$ (1.5M variables) showing that $\varepsilon = 0.2, \lambda = 0$ is a robust default; (c) 24 non-uniform cost experiments confirming no performance degradation.
> > >
> > > We confirm that Sinkhorn convergence speed scales inversely with $\varepsilon$, consistent with the extensive entropic regularization and optimal transport literature in ML. Advances in scalable Sinkhorn solvers from the ML community can directly benefit the pipeline.
> > >
> > > Every solution across all 460+ experiments was verified integral. We will include the full scaling table, graphs, power-law fits, sparsity-aware complexity, sensitivity analysis, and non-uniform terrain experiments. We will also provide the code for full reproducibility.
> > >
> > > We thank the reviewers for their constructive engagement. The questions raised helped us improve the content, and the experimental pipeline we have built in response (over 460 new experiments across scaling, sensitivity, non-uniform costs, and baseline comparisons, all on a single 2022 MacBook) has significantly improved this work.

---

### Decision · Program_Chairs · 2026-04-30

**Decision:**

Accept (spotlight)

**Comment:**

This paper considers the anonymous multi-agent path finding (MAPF) problem.  In plain words, the goal is to send robots to certain target locations by designing the trajectory in a way so that they do not collide in space-time.  Anonymous means that any robot can go to any target.

The main idea in this paper is to cast the MAPF task as a multi-marginal optimal transport (MMOT) instance with some underlying Markovian structure.  The authors show: (i) The OT formulation can be cast as a poly-sized LP with unimodular constraints (in particular, a consequence is that one can choose an optimal solution that is integral), (ii) introduces entropic regularization of the MMOT problem for computational reasons, and (iii) suggest computational improvements using shadow transport.

The consensus among the reviewers is that the connection between MAPF and MMOT is surprising and interesting.  The result on integrality is relevant and important -- one needs maps rather than plans because maps tell us the trajectory of a robot to the target.  This justifies the use of the MMOT formulation.  The entropic regularization facilitates a Sinkhorn-type algorithm, these ideas are standard but they are relevant because of the scale of the problem.  A technical point is that optimal solutions are no longer integral, and hence a projection step is necessary, which the authors also develop.  In all the consensus is that this paper is a well-developed body of work.

The reviewers do make some suggestions especially concerning experimental evaluations, which the authors may wish to consider when making revisions.